# Where to Diffuse, How to Diffuse, and How to Get Back: Automated Learning for Multivariate Diffusions

**Raghav Singhal**[*,1]**, Mark Goldstein**[*1]**, Rajesh Ranganath**[1,2]
Courant Institute of Mathematical Sciences[1], New York University
Center for Data Science[2], New York University

## Abstract

Diffusion-based generative models (DBGMs) perturb data to a target noise distribution and reverse this process to generate samples. The choice of noising process, or *inference diffusion process*, affects both likelihoods and sample quality. For example, extending the inference process with auxiliary variables leads to improved sample quality. While there are many such multivariate diffusions to explore, each new one requires significant model-specific analysis, hindering rapid prototyping and evaluation. In this work, we study Multivariate Diffusion Models (MDMs). For any number of auxiliary variables, we provide a recipe for maximizing a lower-bound on the MDMs likelihood without requiring any model-specific analysis. We then demonstrate how to parameterize the diffusion for a specified target noise distribution; these two points together enable optimizing the inference diffusion process. Optimizing the diffusion expands easy experimentation from just a few well-known processes to an automatic search over all linear diffusions. To demonstrate these ideas, we introduce two new specific diffusions as well as learn a diffusion process on the MNIST, CIFAR10, and IMAGENET32 datasets. We show learned MDMs match or surpass bits-per-dims (BPDs) relative to fixed choices of diffusions for a given dataset and model architecture.

## 1 Introduction

Diffusion-based generative models (DBGMs) perturb data to a target noise distribution and reverse this process to generate samples. They have achieved impressive performance in image generation, editing, translation (Dhariwal & Nichol, 2021; Nichol & Dhariwal, 2021; Sasaki et al., 2021; Ho et al., 2022), conditional text-to-image tasks (Nichol et al., 2021; Ramesh et al., 2022; Saharia et al., 2022) and music and audio generation (Chen et al., 2020; Kong et al., 2020; Mittal et al., 2021). They are often trained by maximizing a lower bound on the log likelihood, featuring an inference process interpreted as gradually "noising" the data (Sohl-Dickstein et al., 2015; Ho et al., 2020).

The choice of this inference process affects both likelihoods and sample quality. On different datasets and models, different inference processes work better; there is no universal best choice of inference, and the choice matters (Song et al., 2020b).

While some work has improved performance by designing score model architectures (Ho et al., 2020; Kingma et al., 2021; Dhariwal & Nichol, 2021), Dockhorn et al. (2021) instead introduce the critically-damped langevin diffusion (CLD), showing that significant improvements in sample generation can be gained by carefully designing new processes. CLD pairs each data dimension with an auxiliary "velocity" variable and diffuses them jointly using second-order Langevin dynamics.

A natural question: if introducing new diffusions results in dramatic performance gains, why are there only a handful of diffusions (variance-preserving stochastic differential equation (VPSDE), variance exploding (VE), CLD, sub-VPSDE) used in DBGMs? For instance, are there other auxiliary variable diffusions that would lead to improvements like CLD? This avenue seems promising as auxiliary variables have improved other generative models and inferences, such as normalizing flows

---

* Equal Contribution. Correspondence to {`rsinghal,goldstein`} at `nyu.edu`.

(Huang et al., 2020), neural ordinary differential equations (ODEs) (Dupont et al., 2019), hierarchical variational models (Ranganath et al., 2016), ladder variational autoencoder (Sønderby et al., 2016), among others.

Despite its success, CLD also provides evidence that each new process requires significant model-specific analysis. Deriving the evidence lower bound (ELBO) and training algorithm for diffusions is challenging (Huang et al., 2021; Kingma et al., 2021; Song et al., 2021) and is carried out in a case-by-case manner for new diffusions (Campbell et al., 2022). Auxiliary variables seemingly complicate this process further; computing conditionals of the inference process necessitates solving matrix Lyupanov equations (section 3.3). Deriving the inference stationary distribution—which helps the model and inference match—can be intractable. These challenges limit rapid prototyping and evaluation of new inference processes.

Concretely, training a diffusion model requires:

(**R1**): Selecting an inference and model process pair such that the inference process converges to the model prior

(**R2**): Deriving the ELBO for this pair

(**R3**): Estimating the ELBO and its gradients by deriving and computing the inference process' transition kernel

In this work, we introduce Multivariate Diffusion Models (MDMs) and a method for training and evaluating them. MDMs are diffusion-based generative models trained with auxiliary variables. We provide a recipe for training MDMs beyond specific instantiations–like VPSDE and CLD—to all linear inference processes that have a stationary distribution, with any number of auxiliary variables.

First, we bring results from gradient-based MCMC (Ma et al., 2015) to diffusion modeling to construct MDMs that converge to a chosen model prior (**R1**); this tightens the ELBO. Secondly, for any number of auxiliary variables, we derive the MDM ELBO (**R2**). Finally, we show that the transition kernel of linear MDMs, necessary for the ELBO, can be computed automatically and generically, for higher-dimensional auxiliary systems (**R3**).

With these tools, we explore a variety of new inference processes for diffusion-based generative models. We then note that the automatic transitions and fixed stationary distributions facilitate directly learning the inference to maximize the MDM ELBO. Learning turns diffusion model training into a search not only over score models but also inference processes, at no extra derivational cost.

**Methodological Contributions.** In summary, our methodological contributions are:

1. Deriving ELBOs for training and evaluating multivariate diffusion models (MDMs) with auxiliary variables.

2. Showing that the diffusion transition covariance does not need to be manually derived for each new diffusion. We instead demonstrate that a matrix factorization technique, previously unused in diffusion models, can automatically compute the covariance analytically for any linear MDM.

3. Using results from gradient-based Markov chain Monte Carlo (MCMC) to construct MDMs with a complete parameterization of inference processes whose stationary distribution matches the model prior.

4. Combining the above into an algorithm called Automatic Multivariate Diffusion Training (AMDT) that enables training without diffusion-specific derivations. AMDT enables training score models for any linear diffusion, including optimizing the diffusion and score jointly.

To demonstrate these ideas, we develop MDMs with two specific diffusions as well as learned multivariate diffusions. The specific diffusions are accelerated Langevin diffusion (ALDA) (introduced in Mou et al. (2019) as a higher-order scheme for gradient-based MCMC) and an alteration, modified accelerated Langevin diffusion (MALDA). Previously, using these diffusions for generative modeling would require significant model-specific analysis. Instead, AMDT for these diffusions is derivation-free.

**Empirical contributions.** We train MDMs on the MNIST, IMAGENET32 and CIFAR-10 datasets. In the experiments, we show that:

1. Training new and existing fixed diffusions, such as ALDA and MALDA, is easy with the proposed algorithm AMDT.

2. Using AMDT to learn the choice of diffusion for the MDM matches or surpasses the performance of fixed choices of diffusion process; sometimes the learned diffusion and VPSDE do best; other times the learned diffusion and CLD do best.

3. There are new and existing MDMs, trained and evaluated with the MDM ELBO, that account for as much performance improvement over VPSDE as a three-fold increase in score model size for a fixed univariate diffusion.

These findings affirm that the choice of diffusion affects the optimization problem, and that learning the choice bypasses the process of choosing diffusions for each new dataset and score architecture. We additionally show the utility of the MDM ELBO by showing on a dataset that CLD achieves better bits-per-dims (BPDs) than previously reported with the probability flow ODE (Dockhorn et al., 2021).

## 2 SETUP

We present diffusions by starting with the generative model and then describing its likelihood lower bound (Sohl-Dickstein et al., 2015; Huang et al., 2021; Kingma et al., 2021). Diffusions sample from a model prior $\mathbf{z}_0 \sim \pi_\theta$ and then evolve a continuous-time stochastic process $\mathbf{z}_t \in \mathbb{R}^d$:

$$d\mathbf{z} = h_\theta(\mathbf{z}, t)dt + \beta_\theta(t)d\mathbf{B}_t, \quad t \in [0, T] \tag{1}$$

where $\mathbf{B}_t$ is a $d$-dimensionsal Brownian motion. The model is trained so that $\mathbf{z}_T$ approximates the data $\mathbf{x} \sim q_{\text{data}}$.[1] Maximum likelihood training of diffusion models is intractable (Huang et al., 2021; Song et al., 2021; Kingma et al., 2021). Instead, they are trained using a variational lower bound on $\log p_\theta(\mathbf{z}_T = x)$. The bound requires an inference process $q_\phi(\mathbf{y}_s | \mathbf{x} = x)$:[2]

$$d\mathbf{y} = f_\phi(\mathbf{y}, s)ds + g_\phi(s)d\widehat{\mathbf{B}}_s, \quad s \in [0, T] \tag{2}$$

where $\widehat{\mathbf{B}}_s$ is another Brownian motion independent of $\mathbf{B}_t$. The inference process is usually taken to be specified rather than learned, and chosen to be i.i.d. for each $y_{tj}$ conditional on each $x_j$. This leads to the interpretation of the $y_{tj}$ as noisy versions of features $x_j$ (Ho et al., 2020). While the diffusion ELBO is challenging to derive in general, Huang et al. (2021); Song et al. (2021) show that when the model process takes the form:

$$d\mathbf{z} = \left[ g_\phi^2(T - t)s_\theta(\mathbf{z}, T - t) - f_\phi(\mathbf{z}, T - t) \right] dt + g_\phi(T - t)d\mathbf{B}_t, \tag{3}$$

the ELBO is:

$$\log p_\theta(x) \geq \mathcal{L}^{\text{ism}}(x) = \mathbb{E}_{q_\phi(\mathbf{y}|x)} \left[ \log \pi_\theta(\mathbf{y}_T) + \int_0^T -\frac{1}{2} \|s_\theta\|_{g_\phi^2}^2 - \nabla \cdot (g_\phi^2 s_\theta - f_\phi) ds \right], \tag{4}$$

where $f_\phi, g_\phi, s_\theta$ are evaluated at $(\mathbf{y}_s, s)$, $\|\mathbf{x}\|_{\mathbf{A}}^2 = \mathbf{x}^\top \mathbf{A} \mathbf{x}$ and $g^2 = gg^\top$. Equation (4) features the Implicit Score Matching (ISM) loss (Song et al., 2020a), and can be re-written as an ELBO $\mathcal{L}^{\text{dsm}}$ featuring Denoising Score Matching (DSM) (Vincent, 2011; Song et al., 2020b), see appendix F.1.

## 3 A RECIPE FOR MULTIVARIATE DIFFUSION MODELS

As has been shown in prior work (Song et al., 2021; Dockhorn et al., 2021), the choice of diffusion matters. Drawing on principles from previous generative models (section 6), we can consider a wide class of diffusion inference processes by constructing them using auxiliary variables.

---

[1]Following Huang et al. (2021); Dockhorn et al. (2021) we integrate all processes in forward time 0 to $T$. It may be helpful to think of an additional variable $\hat{\mathbf{x}}_t \triangleq \mathbf{z}_{T-t}$ so that $\hat{\mathbf{x}}_0$ approximates $\mathbf{x} \sim q_{\text{data}}$.

[2]We use $\mathbf{y}$ as the inference variable over the same space as the model's $\mathbf{z}$.

At first glance, training such diffusions can seem challenging. First, one needs an ELBO that includes auxiliary variables. This ELBO will require sampling from the transition kernel, and setting the model prior to the specified inference stationary distribution. But doing such diffusion-specific analysis manually is challenging and hinders rapid prototyping.

In this section we show how to address these challenges and introduce an algorithm, AMDT, to simplify and automate modeling with MDMs. AMDT can be used to train new and existing diffusions, including those with auxiliary variables, and including those that learn the inference process. In appendix A we discuss how the presented methods can also be used to automate and improve simplified score matching and noise prediction objectives used to train diffusion models.

### 3.1 Multivariate Model and Inference

For the $j^{th}$ data coordinate at each time $t$, MDMs pair $\mathbf{z}_{tj} \in \mathbb{R}$ with a vector of auxiliary variables $\mathbf{v}_{tj} \in \mathbb{R}^{K-1}$ into a joint vector $\mathbf{u}_t$ and diffuse in the extended space:

$$\mathbf{u}_0 \sim \pi_\theta, \qquad d\mathbf{u} = h_\theta(\mathbf{u}_t = \begin{bmatrix} \mathbf{z}_t \\ \mathbf{v}_t \end{bmatrix}, t)dt + \beta_\theta(t)d\mathbf{B}_t. \tag{5}$$

MDMs model the data $\mathbf{x}$ with $\mathbf{z}_T$, a coordinate in $\mathbf{u}_T \sim p_\theta$. For the $j^{th}$ feature $\mathbf{x}_j$, each $\mathbf{u}_{tj} \in \mathbb{R}^K$ consists of a "data" dimension $\mathbf{u}_{tj}^z$ and auxiliary variable $\mathbf{u}_{tj}^v$. Therefore $\mathbf{u} \in \mathbb{R}^{dK}$. We extend the drift coefficient $h_\theta$ from a function in $\mathbb{R}^d \times \mathbb{R}_+ \to \mathbb{R}^d$ to the extended space $\mathbb{R}^{dK} \times \mathbb{R}_+ \to \mathbb{R}^{dK}$. We likewise extend the diffusion coefficient to a matrix $\beta_\theta$ acting on Brownian motion $\mathbf{B}_t \in \mathbb{R}^{dK}$.

Because the MDM model is over the extended space, the inference distribution $\mathbf{y}$ must be too. We then set $q(\mathbf{y}_0^v | \mathbf{y}_0^z = x)$ to any chosen initial distribution, e.g. $\mathcal{N}(\mathbf{0}, \mathbf{I})$ and discuss this choice in section 4. Then $\mathbf{y}_s$ evolves according to the auxiliary variable inference process:

$$d\mathbf{y} = f_\phi(\mathbf{y}, s)ds + g_\phi(s)d\widehat{\mathbf{B}}_s, \tag{6}$$

where the inference drift and diffusion coefficients $f_\phi, g_\phi$ are now over the extended space $\mathbf{y} = [\mathbf{y}^z, \mathbf{y}^v]$. The function $f_\phi$ lets the $z$ and $v$ coordinates of $\mathbf{y}_{tj}$ interact in the inference process.

### Assumptions

This work demonstrates how to parameterize time-varying Itô processes, used for diffusion modeling, to have a stationary distribution that matches the given model prior. To take advantage of the automatic transition kernels also presented, the inferences considered for modeling are linear time-varying processes and take the form:

$$d\mathbf{y} = \mathbf{A}_\phi(s)\mathbf{y}ds + g_\phi(s)d\mathbf{B}_s$$

where $\mathbf{A}_\phi(s) : \mathbb{R}_+ \to dK \times dK$ and $g_\phi(s) : \mathbb{R}_+ \to dK \times dK$ are matrix-valued functions.

### 3.2 ELBO for MDMs

We now show how to train MDMs to optimize a lower bound on the log likelihood of the data. Like in the univariate case, we use the parameterization in eq. (3) to obtain a tractable ELBO.

**Theorem 1.** *The* MDM *log marginal likelihood of the data is lower-bounded by:*

$$\log p_\theta(x) \geq \mathbb{E}_{q_\phi(\mathbf{y}|x)}\left[\underbrace{\log \pi_\theta(\mathbf{y}_T)}_{\ell_T} - \int_0^T \frac{1}{2}\|s_\theta\|_{g_\phi^2}^2 + \nabla \cdot (g_\phi^2 s_\theta - f_\phi)ds - \underbrace{\log q_\phi(\mathbf{y}_0^v|x)}_{\ell_q}\right] \quad (\mathcal{L}^{mism})$$

$$= \mathbb{E}_{q_\phi(\mathbf{y}|x)}\left[\ell_T + \int_0^T \frac{1}{2}\|s_\phi\|_{g_\phi^2}^2 - \frac{1}{2}\|s_\theta - s_\phi\|_{g_\phi^2}^2 + (\nabla \cdot f_\phi)ds - \ell_q\right] \quad (\mathcal{L}^{mdsm}).$$

$$\tag{7}$$

*where divergences and gradients are taken with respect to $\mathbf{y}_s$ and $s_\phi = \nabla_{\mathbf{y}_s} \log q_\phi(\mathbf{y}_s|x)$.*

*Proof.* The proof for the MDM ISM ELBO $\mathcal{L}^{\text{mism}}$ is in appendix F. In short, we introduce auxiliary variables, apply Theorem 1 of Huang et al. (2021) (equivalently, Theorem 3 of Song et al. (2021) or appendix E of Kingma et al. (2021)) to the joint space, and then apply an additional variational bound to $\mathbf{v}_0$. The MDM DSM ELBO $\mathcal{L}^{\text{mdsm}}$ is likewise derived in appendix F, similarly to Huang et al. (2021); Song et al. (2021), but extended to multivariate diffusions. $\square$

We train MDM's by estimating the gradients of $\mathcal{L}^{\text{mdsm}}$, as estimates of $\mathcal{L}^{\text{mism}}$ can be computationally prohibitive. For numerical stability, the integral in eq. (7) is computed on $[\epsilon, T]$ rather than $[0, T]$. One can regard this as a bound for a variable $\mathbf{u}_\epsilon$. To maintain a proper likelihood bound for the data, one can choose a likelihood $\mathbf{u}_0 | \mathbf{u}_\epsilon$ and compose bounds as we demonstrate in appendix I. We report the ELBO with this likelihood term, which plays the same role as the discretized Gaussian in Nichol & Dhariwal (2021) and Tweedie's formula in Song et al. (2021).

## 3.3 INGREDIENT 1: COMPUTING THE TRANSITION $q_\phi(\mathbf{y}_s | x)$

To estimate eq. (7) and its gradients, we need samples from $q(\mathbf{y}_s | x)$ and to compute $\nabla \log q(\mathbf{y}_s | x)$. While an intractable problem for MDMs in general, we provide two ingredients for tightening and optimizing these bounds in a generic fashion for linear inference MDMs.

We first show how to automate computation of $q(\mathbf{y}_s | \mathbf{y}_0)$ and then $q(\mathbf{y}_s | x)$. For linear MDMs of the form:

$$d\mathbf{y} = \mathbf{A}(s)\mathbf{y}ds + g(s)d\mathbf{B}_s,$$

the transition kernel $q(\mathbf{y}_s | \mathbf{y}_0)$ is Gaussian (Särkkä & Solin, 2019). Let $f(\mathbf{y}, s) = \mathbf{A}(s)\mathbf{y}$. Then, the mean and covariance are solutions to the following ODEs:

$$d\mathbf{m}_{s|0}/ds = \mathbf{A}(s)\mathbf{m}_{s|0}$$

$$d\mathbf{\Sigma}_{s|0}/ds = \mathbf{A}(s)\mathbf{\Sigma}_{s|0} + \mathbf{\Sigma}_{s|0}\mathbf{A}^\top(s) + g^2(s). \tag{8}$$

The mean can be solved analytically:

$$\mathbf{m}_{s|0} = \exp\left[\int_0^s \mathbf{A}(\nu)d\nu\right]\mathbf{y}_0 \underbrace{= \exp(s\mathbf{A})\mathbf{y}_0}_{\text{no integration if } \mathbf{A}(\nu) = \mathbf{A}}. \tag{9}$$

The covariance equation does not have as simple a solution because eq. (9) as the unknown matrix $\mathbf{\Sigma}_{s|0}$ is being multiplied both from the left and the right.

Instead of solving eq. (8) for a specific diffusion *manually*, as done in previous work (e.g. pages 50-54 of Dockhorn et al. (2021)), we show that a matrix factorization technique (Särkkä & Solin (2019), sec. 6.3) previously unused in diffusion-based generative models can automatically compute $\mathbf{\Sigma}_{s|0}$ generically for any linear MDM. Define $\mathbf{C}_s, \mathbf{H}_s$ that evolve according to:

$$\begin{pmatrix} d\mathbf{C}_s/ds \\ d\mathbf{H}_s/ds \end{pmatrix} = \begin{pmatrix} \mathbf{A}(s) & g^2(s) \\ \mathbf{0} & -\mathbf{A}^\top(s) \end{pmatrix} \begin{pmatrix} \mathbf{C}_s \\ \mathbf{H}_s \end{pmatrix}, \tag{10}$$

then $\mathbf{\Sigma}_{s|0} = \mathbf{C}_s\mathbf{H}_s^{-1}$ for $\mathbf{C}_0 = \mathbf{\Sigma}_0$ and $\mathbf{H}_0 = \mathbf{I}$ (Appendix D). These equations can be solved in closed-form,

$$\begin{pmatrix} \mathbf{C}_s \\ \mathbf{H}_s \end{pmatrix} = \exp\left[\begin{pmatrix} [\mathbf{A}]_s & [g^2]_s \\ \mathbf{0} & -[\mathbf{A}^\top]_s \end{pmatrix}\right] \begin{pmatrix} \mathbf{\Sigma}_0 \\ \mathbf{I} \end{pmatrix} \underbrace{= \exp\left[s\begin{pmatrix} \mathbf{A} & g^2 \\ \mathbf{0} & -\mathbf{A}^\top \end{pmatrix}\right]}_{\text{no integration if } \mathbf{A}(\nu) = \mathbf{A}, g(\nu) = g} \begin{pmatrix} \mathbf{\Sigma}_0 \\ \mathbf{I} \end{pmatrix}, \tag{11}$$

where $[\mathbf{A}]_s = \int_0^s \mathbf{A}(\nu)d\nu$. To condition on $\mathbf{y}_0 = (x, v)$, we set $\mathbf{\Sigma}_0 = \mathbf{0}$.

**Computing $q_\phi(\mathbf{y}_s | x)$.** For the covariance $\mathbf{\Sigma}_{s|0}$, to condition on $x$ instead of $\mathbf{y}_0$, we set $\mathbf{\Sigma}_0$ to

$$\mathbf{\Sigma}_0 = \begin{pmatrix} 0 & 0 \\ 0 & \mathbf{\Sigma}_{\mathbf{v}_0} \end{pmatrix},$$

To compute the mean, it is the same expression as for $q(\mathbf{y}_s | \mathbf{y}_0)$, but with a different initial condition:

$$\mathbf{m}_{s|0} = \exp\left[\int_0^s \mathbf{A}(\nu)d\nu\right] \begin{pmatrix} x \\ \mathbb{E}_q[\mathbf{y}_0^v | x] \end{pmatrix} \tag{12}$$

See appendix D for more details.

---

**Algorithm 1** Automatic Multivariate Diffusion Training

---

**Input:** Data $\{x_i\}$, inference process matrices $\mathbf{Q}_\phi, \mathbf{D}_\phi$, model prior $\pi_\theta$, initial distribution $q_\phi(\mathbf{y}_0^v \mid x)$, and score model architecture $s_\theta$
**Returns:** Trained score model $s_\theta$
**while** $s_\theta$ not converged **do**
    Sample $x \sim \sum_{i=1}^N \frac{1}{N} \delta_{x_i}$, $v_0 \sim q_\phi(\mathbf{y}_0^v \mid x)$
    Sample $\mathbf{s} \sim \mathbf{U}[0, T]$ and $\mathbf{y}_s, \mathbf{y}_T \sim q_\phi(\mathbf{y}_s \mid x)$ using algorithm 2
    Estimate the stochastic gradient of the MDM ELBO, $\nabla_\theta \mathcal{L}(\theta, \phi)$, using eq. (7)
    $\theta \leftarrow \theta + \alpha \nabla_\theta \mathcal{L}(\theta, \phi)$.
    **if** learning inference **then**
        $\phi \leftarrow \phi + \alpha \nabla_\phi \mathcal{L}(\theta, \phi)$
    **end if**
**end while**
**Output** $s_\theta$

---

**A fast and simple algorithm.** We show in algorithm 2 (appendix H) that computing the transition kernel only requires knowing $f, g$ and requires no diffusion-specific analysis. For $K - 1$ auxiliary variables, $\mathbf{A}, g$ are $K \times K$. Like for scalar diffusions, these parameters are shared across data coordinates. This means matrix exponentials and inverses are done on $K \times K$ matrices, where $K$ is only 2 or 3 in our experiments. In table 1, we compare the time to sample a batch of size 256 from the transition kernel for CIFAR10 and MNIST. The table shows the extra computa-

**Table 1: Runtime Comparison**: we compare the run time of sampling from the CLD diffusion analytically versus using the automated algorithm.

| Method | CIFAR-10 | MNIST |
|---|---|---|
| Analytical | 0.027 | 0.0062 |
| Automated | 0.029 | 0.007 |

tional cost of the automated algorithm is negligible. This automation likewise applies to simplified score matching and noise prediction objectives, since all rely on $q_\phi(\mathbf{y}_s|x)$ (appendix A).

## 3.4 INGREDIENT 2: MDM PARAMETERIZATION

The MDM ELBO (eq. (7)) is tighter when the inference $\mathbf{y}_T$ tends toward the model's prior $\pi_\theta$. Here we construct inference processes with the model prior $\pi_\theta$ as a specified stationary distribution $q_\infty$.

Ma et al. (2015) provide a complete recipe for constructing gradient-based MCMC samplers; the recipe constructs non-linear time-homogeneous Itô processes with a given stationary distribution, and show that the parameterization spans all such Itô processes with that stationary distribution.

Diffusion models usually have time-varying drift and diffusion coefficients (e.g. use of the $\beta(t)$ function). To build diffusion models that match the model prior, we first extend Theorem 1 from Ma et al. (2015) to construct non-linear Itô processes with time-varying drift and diffusion coefficients with a given stationary distribution (Appendix C). Then, to keep transitions tractable (per Section 3.3), we specialize this result to linear Itô diffusions.

We directly state the result for linear time-varying diffusions with stationary distributions. The parameterization requires a skew-symmetric matrix $-\mathbf{Q}(s) = \mathbf{Q}(s)^\top$, a positive semi-definite matrix $\mathbf{D}(s)$, and a function $\nabla H(\mathbf{y})$ such that the desired stationary distribution $q_\infty$ is proportional to $\exp[-H(\mathbf{y})]$. Linear Itô diffusions have Gaussian stationary distributions (Särkkä & Solin, 2019) meaning that $\nabla H$ is linear and can be expressed as $\mathbf{S}\mathbf{y}$ for some matrix $\mathbf{S}$. For a matrix $\mathbf{A}$, let $\sqrt{\mathbf{A}}$ refer to the matrix square root defined by $\mathbf{a} = \sqrt{\mathbf{A}} \iff \mathbf{A} = \mathbf{a}\mathbf{a}^\top$. Then, the Itô diffusion:

$$d\mathbf{y} = \underbrace{-\Big[\mathbf{Q}(s) + \mathbf{D}(s)\Big]\mathbf{S}\mathbf{y}\, ds}_{f(\mathbf{y},s)} + \underbrace{\sqrt{2\mathbf{D}(s)}\, d\widehat{\mathbf{B}}_s}_{g(s)}, \tag{13}$$

has Gaussian stationary distribution $\mathcal{N}(\mathbf{0}, \mathbf{S}^{-1})$ where $\mathbf{Q}(s), \mathbf{D}(s)$ and $\mathbf{S}$ are parameters. For a discussion of convergence to the stationary distribution, as well as skew-symmetric and positive semi-definite parameterizations, see appendix C, where we also show that existing diffusion processes such as VPSDE and CLD are included in $\mathbf{Q}/\mathbf{D}$ parameterization. We display the ELBO in terms of $\mathbf{Q}/\mathbf{D}$ in appendix G and an algorithm in appendix H.

For score matching and noise prediction losses and a given $q_\phi$, achieving a minimizing value with respect to $s_\theta$ does not imply that the generative model score will match the inference score. Modeling the data also requires the marginal distribution of $q_{\phi,T}$ to approximate $\pi$. When $q_\phi$ is constant, it is important to confirm the stationary distribution is appropriately set, and the tools used here for the ELBO can be used to satisfy this requirement for score matching and noise prediction (appendix A).

### 3.5 LEARNING THE INFERENCE PROCESS

The choice of diffusion matters, and the ELBOs in eq. (7) have no requirement for fixed $q_\phi$. We therefore learn the inference process jointly with $s_\theta$. Under linear transitions (ingredient 1), no algorithmic details change as the diffusion changes during training. Under stationary parameterization (ingredient 2), we can learn without the stationary distribution going awry. In the experiments, learning matches or surpasses BPDs of fixed diffusions for a given dataset and score architecture.

In $\mathcal{L}^{\text{mdsm}}$ or $\mathcal{L}^{\text{mism}}$, $q_{\phi,\infty}$ may be set to equal $\pi_\theta$, but it is $\mathbf{y}_T \sim q_{\phi,T}$ for the chosen $T$ that is featured in the ELBO. Learning $q_\phi$ can choose $\mathbf{y}_T$ to reduce the cross-entropy:

$$-\mathbb{E}_{q_\phi(\mathbf{y}_T|x)}[\log \pi_\theta(\mathbf{y}_T)]. \tag{14}$$

Minimizing eq. (14) will tighten the ELBO for any $s_\theta$. Next, $q_\phi$ is featured in the remaining terms that feature $s_\theta$; optimizing for $q_\phi$ will tighten and improve the ELBO alongside $s_\theta$. Finally, $q_\phi$ is featured in the expectations and the $-\log q_\phi$ term:

$$\log p_\theta(\mathbf{u}_T^z = x) \geq = \underbrace{\mathbb{E}_{q_\phi(\mathbf{y}_0^v = v|x)}}\left[(\mathcal{L}^{\text{dsm}} \text{ or } \mathcal{L}^{\text{ism}}) \underbrace{-\log q_\phi(\mathbf{y}_0^v = v|x)}\right] \tag{15}$$

The $q_\phi(\mathbf{y}_0^v|x)$ terms impose an optimality condition that $p_\theta(\mathbf{u}_T^v|\mathbf{u}_T^z) = q_\phi(\mathbf{y}_0^v|\mathbf{y}_0^z)$ (appendix E), When it is satisfied, no looseness in the ELBO is due to the initial time zero auxiliary variables.

To learn, $\mathbf{Q}, \mathbf{D}$ need to be specified with parameters $\phi$ that enable gradients. We keep $\mathbf{S}$ fixed at inverse covariance of $\pi_\theta$. The transition kernel $q_\phi(\mathbf{y}_s|x)$ depends on $\mathbf{Q}, \mathbf{D}$ through its mean and covariance. Gaussian distributions permit gradient estimation with reparameterization or score-function gradients (Kingma & Welling, 2013; Ranganath et al., 2014; Rezende & Mohamed, 2015; Titsias & Lázaro-Gredilla, 2014). Reparameterization is accomplished via:

$$\mathbf{y}_s = \mathbf{m}_{s|0} + \mathbf{L}_{s|0}\epsilon \tag{16}$$

where $\epsilon \sim \mathcal{N}(0, I_{dK})$ and $\mathbf{L}_{s|0}$ satisfies $\mathbf{L}_{s|0}\mathbf{L}_{s|0}^\top = \mathbf{\Sigma}_{s|0}$, derived using coordinate-wise Cholesky decomposition. Gradients flow through eq. (16) from $\mathbf{y}_s$ to $\mathbf{m}_{s|0}$ and $\mathbf{\Sigma}_{s|0}$ to $\mathbf{Q}, \mathbf{D}$ to parameters $\phi$.

Algorithm 1 displays Automatic Multivariate Diffusion Training (AMDT). AMDT provides a training method for diffusion-based generative models for either fixed $\mathbf{Q}, \mathbf{D}$ matrices or for learning the $\mathbf{Q}_\phi, \mathbf{D}_\phi$ matrices, without requiring any diffusion-specific analysis.

**Learning in other diffusion objectives.** Like in the ELBO, learning in score matching or noise prediction objectives can improve the match between the inference process and implied generative model (appendix A).

## 4 INSIGHTS INTO MULTIVARIATE DIFFUSIONS

**Scalar versus Multivariate Processes.** Equation (13) clarifies what can change while preserving $q_\infty$. Recall that $\mathbf{Q}$ and $\mathbf{D}$ are $K \times K$ for $K - 1$ auxiliary variables. Because $0$ is the only $1 \times 1$ skew-symmetric matrix, scalar processes must set $\mathbf{Q} = 0$. With $q_{\phi,\infty} = \mathcal{N}(0, \mathbf{I})$, the process is:

$$d\mathbf{y} = -\mathbf{D}(s)\mathbf{y}ds + \sqrt{2\mathbf{D}(s)}d\widehat{\mathbf{B}}_s. \tag{17}$$

What is left is the VPSDE process used widely in diffusion models where $\mathbf{D}(s) = \frac{1}{2}\beta(s)$ is $1 \times 1$ (Song et al., 2020b). This reveals that the VPSDE process is the only scalar diffusion with a stationary distribution.[3] This also clarifies the role of $\mathbf{Q}$: it accounts for mixing between dimensions in multivariate processes, as do non-diagonal entries in $\mathbf{D}$ for $K > 1$.

---

[3]There are processes such as sub-VPSDE (Song et al., 2020b) which are covered in the sense that they tend to members of this parameterization as $T$ grows: sub-VP converges to VPSDE.

**CLD optimizes a log-likelihood lower bound.** Differentiating $\mathcal{L}^{\text{mdsm}}$ (eq. (7)) with respect to the score model parameters, we show that the objective for CLD (Dockhorn et al., 2021) maximizes a lower bound on $\log p_\theta(x)$, not just $\log p_\theta(\mathbf{u}_0)$, without appealing to the probability flow ODE.

**Does my model use auxiliary variables?** An example initial distribution is $q(\mathbf{y}_0^v|x) = \mathcal{N}(0, \mathbf{I})$. It is also common to set $\pi_\theta = \mathcal{N}(0, \mathbf{I})$. Because the optimum for diffusions is $p_\theta = q$, the optimal model has main and auxiliary dimensions independent at endpoints $0$ and $T$. Does this mean that the model does not use auxiliary variables? In appendix B, we show that in this case the model can still use auxiliary variables at intermediate times. A sufficient condition is non-diagonal $\mathbf{Q} + \mathbf{D}$.

## 5 EXPERIMENTS

We test the MDM framework with handcrafted and learned diffusions. The handcrafted diffusions are (a) ALDA, used in (Mou et al., 2019) for accelerated gradient-based MCMC sampling (eq. (32)) and (b) MALDA: a modified version of ALDA (eq. (33)). Both have two auxiliary variables. We also learn diffusions with 1 and 2 auxiliary variables. We compare with VPSDE and ELBO-trained CLD.

**Table 2:** BPD upper-bounds on image generation for a fixed architecture. CIFAR-10: learning outperforms CLD, and both outperform the standard choice of VPSDE. MNIST: learning matches VPSDE while the fixed auxiliary diffusions are worse. IMAGENET32: all perform similarly. Learning matches or surpasses the best fixed diffusion, while bypassing the need to choose a diffusion.

| Model | $K$ | CIFAR-10 | IMAGENET32 | MNIST |
|---|---|---|---|---|
| VPSDE | 1 | 3.20 | 3.70 | 1.26 |
| Learned | 2 | 3.07 | 3.71 | 1.28 |
| Learned | 3 | 3.08 | 3.72 | 1.33 |
| CLD | 2 | 3.11 | 3.70 | 1.35 |
| MALDA | 3 | 3.13 | 3.72 | 1.65 |
| ALDA | 3 | 29.43 | 33.08 | 124.60 |

**Table 3:** Parameter Efficiency. The first two rows display diffusions from previous work: VPSDE and CLD, both using score models with **108 million** parameters on CIFAR-10. We train the rest using a score model with **35.7 million** parameters. The learned diffusion matches the performance of VPSDE-large; changes in the inference can account for as much improvement as a 3x increase in score parameters. BPDs are upper-bounds.

| Model | $K$ | Parameters | CIFAR-10 |
|---|---|---|---|
| VPSDE-large (Song et al., 2021) | 1 | 108M | 3.08 |
| CLD-large (Dockhorn et al., 2021) | 2 | 108M | 3.31 |
| Learned | 2 | 35.7M | 3.07 |
| CLD | 2 | 35.7M | 3.11 |
| VPSDE | 1 | 35.7M | 3.20 |

Following prior work, we train DBGMs for image generation. We use the U-Net from Ho et al. (2020). We input the auxiliary variables as extra channels, which only increases the score model parameters in the input and output convolutions (CLD and Learned 2 have $7,000$ more parameters than VPSDE on CIFAR-10 and IMAGENET32 and only $865$ more for MNIST). We use simple uniform dequantization. We report estimates of $\mathcal{L}^{\text{mdsm}}$ (which reduces to the standard $\mathcal{L}^{\text{dsm}}$ for $K = 1$). We sample times using the importance sampling distribution from Song et al. (2021) with truncation set to $\epsilon = 10^{-3}$. To ensure the truncated bound is proper, we use a likelihood described in appendix I.

**Results.** Table 2 shows that the inference process matters and displays. It displays DBGMs that we train and evaluate on CIFAR-10, IMAGENET32 and MNIST. This includes the existing VPSDE and CLD, the new MALDA and ALDA, and the new learned inference processes. All are trained with the 35.7M parameter architecture. For CIFAR-10, learning outperforms CLD, and both outperform the standard choice of VPSDE. For MNIST, learned diffusions match VPSDE while the three fixed auxiliary diffusions are worse. On IMAGENET32, all perform similarly. The take-away is that learning

matches or surpasses the best fixed diffusion performance and bypasses the choice of diffusion for each new dataset or score architecture. In Figure 1 we plot the generated samples from CIFAR10.

Table 3's first two rows display diffusion models from previous work: VPSDE (Song et al., 2021) and CLD (Dockhorn et al., 2021) both with the **108 million** score model from Song et al. (2021) (labeled "large"). The rest are DBGMs that we train using the U-Net with **35.7 million** parameters for CIFAR-10 and IMAGENET32 and 1.1 million for MNIST. Despite using significantly fewer parameters, the learned diffusion achieves similar BPD compared to the larger models, showing that changes in inference can account for as much improvement as a three-fold increase in parameters. While the larger architecture requires two GPUs for batch size 128 on CIFAR-10 on A100s, the smaller one only requires one; exploring inference processes can make diffusions more computationally accessible. Table 3 also demonstrates a tighter bound for CLD trained and evaluated with the MDM ELBO ($\leq 3.11$) relative to existing probability flow-based evaluations (3.31).

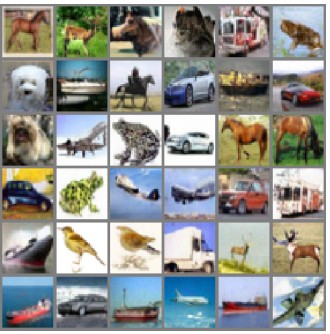 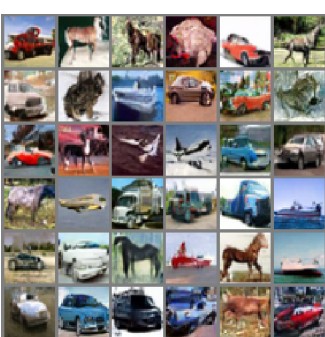

**Figure 1:** CIFAR10 samples generated from the "learned 2" and MALDA generative models.

## 6 RELATED WORK

**Evidence Lower Bounds.** Song et al. (2021); Huang et al. (2021) derive the ISM and DSM lower-bounds on the model log likelihood. Our work extends their analysis to the multivariate diffusion setting to derive lower bounds on the log marginal of the data in the presence of auxiliary variables.

**Auxiliary variables.** Dupont et al. (2019) shows that augmented neural ODEs model a richer set of functions and Huang et al. (2020) uses this principle for normalizing flows. Hierarchical variational models and auto-encoders marginalize auxiliary variables to build expressive distributions (Ranganath et al., 2016; Sønderby et al., 2016; Maaløe et al., 2019; Vahdat & Kautz, 2020; Child, 2020). We apply this principle to DBGMs, including and extending CLD (Dockhorn et al., 2021).

**Learning inference.** Learning $q_\phi$ with $p_\theta$ is motivated in previous work (Kingma & Welling, 2013; Sohl-Dickstein et al., 2015; Kingma et al., 2021). Kingma et al. (2021) learn the noise schedule for VPSDE. For MDMs, there are parameters to learn beyond the noise schedule; $\mathbf{Q}$ can be non-zero, $\mathbf{D}$ can diagonal or full, give $\mathbf{Q}$ and $\mathbf{D}$ different time-varying functions, and learn $\nabla \mathbf{H}$.

## 7 DISCUSSION

We present an algorithm for training multivariate diffusions with linear time-varying inference processes with a specified stationary distribution and any number of auxiliary variables. This includes automating transition kernel computation and providing a parameterization of diffusions that have a specified stationary distribution, which facilitate working with new diffusion processes, including learning the diffusion. The experiments show that learning matches or surpasses the best fixed diffusion performance, bypassing the need to choose a diffusion. MDMs achieve BPDs similar to univariate diffusions, with as many as three times more score parameters. The proposed MDM ELBO reports a tighter bound for the existing CLD relative to existing probability flow-based evaluations. This work enables future directions including interactions across data coordinates and using new stationary distributions.

## 8 ACKNOWLEDGEMENTS

This work was generously funded by NIH/NHLBI Award R01HL148248, NSF Award 1922658 NRT-HDR: FUTURE Foundations, Translation, and Responsibility for Data Science, and NSF CAREER Award 2145542. The authors would additionally like to thank Chin-Wei Huang for helpful discussing regarding Huang et al. (2021).

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

## A    AUTOMATED SCORE MATCHING WITH LEARNED INFERENCE

Like for the MDM ELBO, the methods in this work apply to training with the score matching loss:

$$\mathcal{L}_{\text{SM}}(x, \theta, \phi) = T\mathbb{E}_{t \sim U[0,T]}\mathbb{E}_{q_\phi(\mathbf{y}|x)} \left[ \lambda(t) \left\| s_\theta(\mathbf{y}_t, t) - \nabla_{\mathbf{y}_t} \log q_\phi(\mathbf{y}_t \mid x) \right\|_2^2 \right],$$

where $\lambda : [0, T] \to \mathbb{R}_+$ is a weighing function. The score-matching loss is often optimized in its simplified noise prediction form:

$$\mathcal{L}_{\text{NP}}(x, \theta, \phi) = T\mathbb{E}_{t \sim U[0,T]}\mathbb{E}_{q_\phi(\mathbf{y}|x)} \left[ \left\| \epsilon_\theta(\mathbf{y}_t, t) - \epsilon \right\|_2^2 \right]$$

where $s_\theta = -\mathbf{L}_t^{-\top}\epsilon_\theta$ and $\mathbf{y}_t = \mu_t + \mathbf{L}_t\epsilon$ and $\epsilon$ is the noise used in sampling $\mathbf{y}_t$. We describe here how the improvements to the ELBO studied in this work carry over to $\mathcal{L}_{\text{SM}}$ and $\mathcal{L}_{\text{NP}}$. In the following let $q_0$ be the data distribution, let $p_{(\theta,\phi),0}$ be the model's distribution of the data, and recall that the model is defined by $(s_\theta, f_\phi, g_\phi)$ and prior $\pi$ via a continuous-time stochastic process with drift coefficient $g_\phi^2 s_\theta - f_\phi$ and and diffusion coefficient $g_\phi$.

First, minimizing $\mathcal{L}_{\text{SM}}$ or $\mathcal{L}_{\text{NP}}$ so that $\nabla_{\mathbf{y}_t} \log q_\phi(\mathbf{y}_t) = s_\theta(\mathbf{y}_t, t)$ does not alone imply that $p_{(\theta,\phi),0}$ will equal $q_0$; it must also be that $q_{\phi,T} \approx \pi$. Foregoing this requirement means $\pi$ will produce samples that the generative model may not be able to push onto the path the model was trained on (formally, the score of the generative model would not equal the time-reversal of the forward score even if $s_\theta$ equals the forward score). This condition can be satisfied if $q_\phi$ can be chosen with stationary distribution $\pi$. Section 3.4 describes how to accomplish this.

Next, for any fixed $q_\phi$, automatic transitions from section 3.3 streamline the computation of the score matching loss, allowing for simple score computation for a wide class of diffusions beyond VP.

Finally, for a fixed $q_\phi$ with $q_{\phi,T} \approx \pi$ and a score architecture $s_\theta$, minimizing $\mathcal{L}_{\text{SM}}$ or $\mathcal{L}_{\text{NP}}$ w.r.t $\theta$ may be suboptimal. Optimization, like for the elbo, carries over to score matching and can close this gap; learning w.r.t. both $\theta, \phi$ increases the ability to successfully minimize the loss at each $t$ (section 3.5). In other words, since the generative model is defined by $(s_\theta, f_\phi, g_\phi)$, learning $q_\phi$ means the loss trains all three components of the generative model rather than just one. In summary, score matching is automatic and can learn over the space of linear diffusions that tend to the model prior.

## B    DOES MY MODEL USE AUXILIARY VARIABLES?

In section 3 we gave the example choice of $q(\mathbf{y}_0^v|x) = \mathcal{N}(0, \mathbf{I})$ coordinate-wise. It is also a common choice to set $\pi_\theta = \mathcal{N}(0, \mathbf{I})$. Because the optimum in diffusion models is $p_\theta = q$ for all $t$, we see a peculiar phenomenon under this choice: the model has main and auxiliary dimensions independent at both endpoints $0$ and $T$. Does this mean that the model does not use auxiliary variables? We show that even when $q_\phi(\mathbf{y}_0)$ and $\pi_\theta$ have main and auxiliary variables independent, the model can use the auxiliary variables. A sufficient condition is $\mathbf{Q} + \mathbf{D}$ is non-diagonal.

To make this precise, we recall that we model with $p_\theta(\mathbf{u}_T^z = x)$. To show the model is using auxiliary variables, we just need to show that $\mathbf{u}_T^z$ (main coordinate at $T$) depends on $\mathbf{u}_t^v$ (aux. coordinate at $t$) for $T > t$. At optimum, $p_\theta(\mathbf{u}_T^z, \mathbf{u}_t^v) = q_\phi(\mathbf{y}_0^z, \mathbf{y}_{T-t}^v)$. Therefore it is sufficient to show that for some time $s$, $q_\phi(\mathbf{y}_s^v|\mathbf{y}_0^z) \neq q_\phi(\mathbf{y}_s^v)$. Because $\mathbf{y}_s^z$, is determined by $x$ we need to show that $q_\phi(\mathbf{y}_s^v|x) \neq q_\phi(\mathbf{y}_s^v)$. To do that, we first derive $q(\mathbf{y}_s|x)$ and then marginalize to get $q(\mathbf{y}_s^v|x)$ from it. Since the former is 2D Gaussian, the latter is available in terms of the former's mean and covariance. Suppose $\mathbb{E}[\mathbf{y}_0^v] = 0$, $\mathbf{Q} = [[0, -1], [1, 0]]$ and $\mathbf{D} = [[1, 0], [0, 1]]$ and we have $s = .1$ We have:

$$\mathbb{E}[\mathbf{y}_s|x] = \exp\left[ -s(\mathbf{Q} + \mathbf{D}) \right]\begin{pmatrix} x \\ 0 \end{pmatrix} = \exp\left[ \begin{bmatrix} -.1 & .1 \\ -.1 & -.1 \end{bmatrix} \right] \begin{pmatrix} x \\ 0 \end{pmatrix} = \begin{pmatrix} 0.9003x \\ -0.090x \end{pmatrix} \quad (18)$$

Regardless of the covariance any 1D of this 2D gaussian will have mean that is a function of $x$, meaning that $q(\mathbf{y}_s^v|x)$ does not equal $q(\mathbf{y}_s^v)$ (which is also a Gaussian but with mean depending on $\mathbf{x}'s$ mean rather than $x$ itself. Therefore, even under the setup with independent endpoints, the optimal model makes use of the intermediate auxiliary variables in its final modeling distribution $p_\theta(\mathbf{u}_T^z = x)$.

Are there choices of $\mathbf{Q}$ and $\mathbf{D}$ that lead to learning models that don't make use of the extra dimensions? As mentioned, in the inference process, $\mathbf{Q}$ is responsible for mixing information among the

coordinates, and is the only source of this when $\mathbf{D}$ is diagonal. Then, if $\mathbf{Q} = \mathbf{0}$ and $\mathbf{D}$ is diagonal, none of the coordinates for a given feature $\mathbf{x}_j$ (including $\mathbf{u}_{tj}^z, \mathbf{u}_{tj}^{v_1}, \ldots, \mathbf{u}_{tj}^{v_{K-1}}$) interact for any $t$. Then, since $p_\theta = q$ at optimum, independence of the coodinates at all $t$ in $q$ imply the same in $p_\theta$ and the model will not make use of any auxiliary variables when modeling the marginal $\log p_\theta(\mathbf{u}_T^z = x)$.

## C   STATIONARY PARAMETERIZATION

The non-linear time-homogeneous Itô process family is:

$$dy = f(\mathbf{y})dt + g(\mathbf{y})\mathbf{B}_t. \tag{19}$$

This family can be restricted to those with stationary distributions. Ma et al. (2015) show a complete recipe to span the subset of this family with a desired stationary distribution. Let $\mathbf{Q}$ be skew-symmetric ($-\mathbf{Q} = \mathbf{Q}^\top$) and $\mathbf{D}$ is positive semi-definite. Suppose the desired stationary distribution is $q_\infty(\mathbf{y})$. For a matrix $\mathbf{A}$, let $\sqrt{\mathbf{A}}$ refer to the matrix square root defined by $\mathbf{a} = \sqrt{\mathbf{A}} \iff \mathbf{A} = \mathbf{aa}^\top$. Then, Ma et al. (2015) show that, setting $\mathbf{H}(\mathbf{y}) = -\log q_\infty(\mathbf{y})$, $g(\mathbf{y}) = \sqrt{2\mathbf{D}(\mathbf{y})}$, and

$$f(\mathbf{y}) = -[\mathbf{D}(\mathbf{y}) + \mathbf{Q}(\mathbf{y})]\nabla\mathbf{H}(\mathbf{y}) + \mathbf{\Gamma}(\mathbf{y}), \qquad \mathbf{\Gamma}_i(\mathbf{y}) = \sum_{j=1}^d \frac{\partial}{\partial \mathbf{z}_j}(\mathbf{D}_{ij}(\mathbf{y}) + \mathbf{Q}_{ij}(\mathbf{y})), \tag{20}$$

yields a process $\mathbf{y}_t$ with stationary distribution $q_\infty$. We extend it to time-varying (time inhomogeneous) processes.

**Theorem 2.** $q_\infty(\mathbf{y}) \propto \exp[-H(\mathbf{y})]$ *is a stationary distribution of*

$$dy = \left(-[\mathbf{D}(\mathbf{y},t) + \mathbf{Q}(\mathbf{y},t)]\nabla\mathbf{H}(\mathbf{y}) + \mathbf{\Gamma}(\mathbf{y},t)\right)dt + \sqrt{2\mathbf{D}(\mathbf{y},t)}\mathbf{B}_t, \tag{21}$$

*for*

$$\mathbf{\Gamma}_i(\mathbf{y},t) = \sum_{j=1}^d \frac{\partial}{\partial \mathbf{y}_j}(\mathbf{D}_{ij}(\mathbf{y},t) + \mathbf{Q}_{ij}(\mathbf{y},t)). \tag{22}$$

*Proof.* The Fokker Planck equation is:

$$\partial_t q(\mathbf{y},t) = -\sum_i \frac{\partial}{\partial \mathbf{y}_i}\Big[f_i(\mathbf{y},t)q(\mathbf{y},t)\Big] + \sum_{i,j} \frac{\partial^2}{\partial \mathbf{y}_i \partial \mathbf{y}_j}\Big[\mathbf{D}_{ij}(\mathbf{y},t)q(\mathbf{y},t)\Big] \tag{23}$$

A stationary distribution is one where the Fokker-Planck right hand side is equal to $0$. To show that the stationary characterization also holds of time-inhomogenous processes with $\mathbf{D}(\mathbf{y},t)$ and $\mathbf{Q}(\mathbf{y},t)$, we take two steps, closely following Yin & Ao (2006); Shi et al. (2012); Ma et al. (2015), but noting that there is no requirement for $\mathbf{Q}, \mathbf{D}$ to be free of $t$. First, we show that the Fokker-Plack equation can be re-written as:

$$\partial_t q(\mathbf{y},t) = \nabla \cdot \left(\Big[\mathbf{D}(\mathbf{y},t) + \mathbf{Q}(\mathbf{y},t)\Big]\Big[q(\mathbf{y},t)\nabla H(\mathbf{y}) + \nabla q(\mathbf{y},t)\Big]\right) \tag{24}$$

Second, because the whole expression is set to $0$ when the inside expression equals $0$

$$q(\mathbf{y},t)\nabla H(\mathbf{y}) + \nabla q(\mathbf{y},t) = 0, \tag{25}$$

we just need to show that this holds when $q(\mathbf{y},t) = \exp[-H(\mathbf{y})]/\mathbf{Z}$. The second step is concluded because

$$\Big[q(\mathbf{y},t)\nabla H(\mathbf{y}) + \nabla q(\mathbf{y},t)\Big] = \frac{1}{\mathbf{Z}}\Big[\exp[-H(\mathbf{y})]\nabla H(\mathbf{y}) + \nabla \exp[-H(\mathbf{y})]\Big] = 0,$$

where $\mathbf{Z}$ is the normalization constant of $\exp(-H(y))$.

It only remains to show that Fokker-Plack can be re-written in divergence form with time-dependent $\mathbf{Q}, \mathbf{D}$. In the following let $Q_{ijt}$ denote $\mathbf{Q}_{ij}(\mathbf{y}, t)$ and likewise for $D_{ijt}$. Let $\partial_i$ denote $\frac{\partial}{\partial \mathbf{y}_i}$ and let it denote $\frac{d}{d\mathbf{y}_i}$ for scalar functions. We will use $[Ax]_i = \sum_j A_{ij}x_j$.

$$\partial_t q_t = \nabla \cdot \left( [\mathbf{D}(\mathbf{y}, t) + \mathbf{Q}(\mathbf{y}, t)][q\nabla H + \nabla q] \right)$$

$$= \sum_i \partial_i \left( \left[ [\mathbf{D}(\mathbf{y}, t) + \mathbf{Q}(\mathbf{y}, t)][q\nabla H + \nabla q] \right]_i \right)$$

$$= \sum_i \partial_i \sum_j [D_{ijt} + Q_{ijt}][q\nabla H + \nabla q]_j$$

$$= \sum_i \partial_i \sum_j [D_{ijt} + Q_{ijt}][q\partial_j H + \partial_j q]$$

$$= \sum_i \partial_i \sum_j [D_{ijt} + Q_{ijt}][q\partial_j H] + \sum_i \partial_i \sum_j [D_{ijt} + Q_{ijt}][\partial_j q]$$

$$= \sum_i \partial_i \sum_j [D_{ijt} + Q_{ijt}][q\partial_j H] + \sum_i \partial_i \sum_j D_{ijt}[\partial_j q] + \sum_i \partial_i \sum_j Q_{ijt}[\partial_j q]$$

We re-write the 2nd and 3rd term. Holding $i$ fixed and noting $q$ is scalar, we get the product rule $\sum_j D_{ijt}(\partial_j q) = \sum_j \partial_j [D_{ijt}q] - q\sum_j \partial_j D_{ijt}$ for each $i$, and likewise for $q$:

$$\sum_i \partial_i \sum_j [D_{ijt} + Q_{ijt}][q\partial_j H] + \sum_i \partial_i \sum_j D_{ijt}[\partial_j q] + \sum_i \partial_i \sum_j Q_{ijt}[\partial_j q]$$

$$= \sum_i \partial_i \sum_j [D_{ijt} + Q_{ijt}][q\partial_j H] + \sum_i \partial_i \sum_j \partial_j [D_{ijt}q] - q\sum_j \partial_j D_{ijt}$$

$$+ \sum_i \partial_i \sum_j \partial_j [Q_{ijt}q] - q\sum_j \partial_j Q_{ijt}$$

Because $\mathbf{Q}(\mathbf{y}, t)$ is skew-symmetric, we have that $\sum_i \partial_i \sum_j \partial_j [Q_{ijt}q] = 0$, leaving

$$\partial_t q_t = \sum_i \partial_i \left[ \sum_j [D_{ijt} + Q_{ijt}][q\partial_j H] \right] + \sum_i \partial_i \left[ \sum_j \partial_j [D_{ijt}q] - q\sum_j \partial_j D_{ijt} - q\sum_j \partial_j Q_{ijt} \right]$$

$$= \sum_i \partial_i \left[ \sum_j [D_{ijt} + Q_{ijt}][\partial_j H]q \right] + \sum_i \partial_i \left[ \sum_j \partial_j [D_{ijt}q] - q\sum_j \partial_j (D_{ijt} + Q_{ijt}) \right]$$

$$= \sum_i \partial_i \left[ \left( \sum_j [D_{ijt} + Q_{ijt}][\partial_j H] - \sum_j \partial_j (D_{ijt} + Q_{ijt}) \right)q \right] + \sum_i \sum_j \frac{\partial^2}{\mathbf{y}_i \mathbf{y}_j}(D_{ijt}q)$$

Recalling that $f_i(\mathbf{y}, t) = \left( -[D + Q]\nabla H + \Gamma \right)_i$ and again that $[Ax]_i = \sum_j A_{ij}x_j$, we have equality with the original Fokker-Planck

$$= \sum_i \partial_i \left[ \left( \sum_j [D_{ijt} + Q_{ijt}][\partial_j H] - \sum_j \partial_j (D_{ijt} + Q_{ijt}) \right)q \right] + \sum_{ij} \frac{\partial^2}{\mathbf{y}_i \mathbf{y}_j}(D_{ijt}q)$$

$$= -\sum_i \frac{\partial}{\partial \mathbf{y}_i} \left[ f_i(\mathbf{y}, t)q(\mathbf{y}, t) \right] + \sum_{ij} \frac{\partial^2}{\mathbf{y}_i \mathbf{y}_j} \left[ \mathbf{D}_{ij}(\mathbf{y}, t)q(\mathbf{y}, t) \right]$$

$$= \partial_t q(\mathbf{y}, t)$$

$\square$

We have shown $\exp[-H(\mathbf{y})]/\mathbf{Z}$ is a stationary distribution of the time-varying non-linear Itô process:

$$d\mathbf{y} = \left( -[\mathbf{D}(\mathbf{y}, t) + \mathbf{Q}(\mathbf{y}, t)]\nabla H(\mathbf{y}) + \mathbf{\Gamma}(\mathbf{y}, t) \right)dt + \sqrt{2\mathbf{D}(\mathbf{y}, t)}\mathbf{B}_t. \tag{26}$$

However, for some choices of $\mathbf{Q}, \mathbf{D}$, $\exp[-H(\mathbf{y})]/\mathbf{Z}$ is not necessarily the unique stationary distribution. One problematic case can occur as follows. Suppose that row $i$ of $(\mathbf{Q} + \mathbf{D})$ is all-zero; in this case, $d\mathbf{y}_i = 0$ which implies that $(\mathbf{y}_i)_t = (\mathbf{y}_i)_0$ for all $t > 0$. Then, the initial distribution is also a stationary distribution. To rule out such pathological diffusions, we make the assumption that $\mathbf{Q} + \mathbf{D}$ is full rank. Then, for uniqueness, recall that stationary distributions are the zeros of

$$\partial_t q(\mathbf{y}, t) = \nabla \cdot \left( \left[ \mathbf{D}(\mathbf{y}, t) + \mathbf{Q}(\mathbf{y}, t) \right] \left[ q(\mathbf{y}, t) \nabla H(\mathbf{y}) + \nabla q(\mathbf{y}, t) \right] \right)$$

where the expression is of the form $\mathbf{A}\mathbf{v}$ for $\mathbf{A} = \mathbf{D}(\mathbf{y}, t) + \mathbf{Q}(\mathbf{y}, t)$ and

$$\mathbf{v} = \left[ q(\mathbf{y}, t) \nabla H(\mathbf{y}) + \nabla q(\mathbf{y}, t) \right].$$

Under the assumption that $\mathbf{Q} + \mathbf{D}$ is full rank, the expression can only be zero when $\mathbf{v}$ is zero. To show uniqueness under the full rank assumption, one must then show that

$$\nabla q(\mathbf{y}, t) = -q(\mathbf{y}, t) \nabla H(\mathbf{y}).$$

holds only if $q(\mathbf{y}, t) = \exp[-H(\mathbf{y})]/\mathbf{Z}$. Even if $\exp[-H(\mathbf{y})]/\mathbf{Z}$ is the unique stationary distribution, convergence to that distribution is a question. See Zhang & Chen (2013) for more details.

Learning $\mathbf{Q}_\phi, \mathbf{D}_\phi$ in the MDM ELBO helps push $\mathbf{y}_T$ to the model prior $\pi_\theta$ and avoid issues like those discussed.

## C.1 LINEAR PROCESSES

Next, we specialize this general family to linear Itô processes to maintain tractable transition distributions. A linear process is one where the drift $f(\mathbf{y}, t)$ and diffusion $g(\mathbf{y}, t)$ are linear functions of $\mathbf{y}$. We express the drift function of a non-linear time-varying Itô process with stationary distribution proportional to $\exp[-H(\mathbf{y})]$ as

$$-(\mathbf{Q}(\mathbf{y}, t) + \mathbf{D}(\mathbf{y}, t)) \nabla H(\mathbf{y}) + \Gamma(\mathbf{y}, t).$$

Next, linear Itô processes have Gaussian stationary distributions (Särkkä & Solin, 2019) so $H(\mathbf{y})$ must be quadratic and $\nabla H(\mathbf{y})$ is linear, and neither are constant in $\mathbf{y}$. Because $\nabla H(\mathbf{y})$ is linear, it can be expressed as $\mathbf{S}\mathbf{y}$ for some matrix $\mathbf{S}$ where $\mathbf{S}$ is the inverse of the covariance matrix. Because $\nabla H$ is multiplied by $\mathbf{Q}, \mathbf{D}$, this means that $\mathbf{Q}, \mathbf{D}$ must be free of $\mathbf{y}$. Recalling that $\Gamma$ is expressed as a sum of derivatives w.r.t $\mathbf{y}$ of $\mathbf{Q} + \mathbf{D}$, this means that $\Gamma$ must satisfy $\Gamma = 0$. Next, because of the stationary requirement that $g(t) = \sqrt{2\mathbf{D}(\mathbf{y}, t)}$, we can also conclude by the restriction on $\mathbf{D}$ that the diffusion coefficient function must be independent of the state $\mathbf{y}$. Our final form for linear time-varying processes with stationary distributions $\mathcal{N}(0, \mathbf{S}^{-1})$ is:

$$d\mathbf{y} = \underbrace{-\left[ \mathbf{Q}(t) + \mathbf{D}(t) \right] \mathbf{S}\mathbf{y}}_{f(\mathbf{y}, t)} \, dt + \underbrace{\sqrt{2\mathbf{D}(t)}}_{g(t)} \, d\mathbf{B}_t \tag{27}$$

## C.2 PARAMETERIZING $\mathbf{Q}_\phi$

Suppose $b_q(s)$ is a positive scalar function defined on the time domain with known integral. Suppose $\tilde{\mathbf{Q}}_\phi$ is any matrix. Then $\tilde{\mathbf{Q}}_\phi - \tilde{\mathbf{Q}}_\phi^\top$ is skew-symmetric with $\tilde{\mathbf{Q}}_{\phi,ij} = -\tilde{\mathbf{Q}}_{\phi,ji}$. We can set $\mathbf{Q}_\phi$ to

$$\mathbf{Q}_\phi(s) = b_q(s) \cdot \left[ \tilde{\mathbf{Q}}_\phi - \tilde{\mathbf{Q}}_\phi^\top \right] \tag{28}$$

This is a general parameterization of time-independent skew-symmetric matrices, which have number of degrees of freedom equal to the number of entries in one of the triangles of the matrix, excluding the diagonal.

## C.3 PARAMETERIZING $\mathbf{D}_\phi$

Suppose $b_d(s)$ is a positive scalar function defined on the time domain with known integral. Suppose $\tilde{\mathbf{D}}_\phi$ is any matrix. Then $\tilde{\mathbf{D}}_\phi \tilde{\mathbf{D}}_\phi^\top$ is positive semi-definite and spans all time-independent positive

semi-definite matrices. We can set $\mathbf{D}_\phi$ to

$$\mathbf{D}_\phi(s) = b_d(s) \cdot \left[ \tilde{\mathbf{D}}_\phi \tilde{\mathbf{D}}_\phi^\top \right] \tag{29}$$

To show $\tilde{\mathbf{D}}\tilde{\mathbf{D}}^\top$ spans all positive semi-definite matrices: suppose $\mathbf{M}$ is positive semi-definite. Then it is square. Then it can be eigen-decomposed into $\mathbf{M} = \mathbf{V}\boldsymbol{\Sigma}\mathbf{V}^\top$ The degrees of freedom in $\mathbf{V}\boldsymbol{\Sigma}\mathbf{V}^\top$ are just $\mathbf{R} = \mathbf{V}\sqrt{\boldsymbol{\Sigma}}$ since $\mathbf{V}\boldsymbol{\Sigma}\mathbf{V}^\top = \mathbf{R}\mathbf{R}^\top$ and the square root is taken element-wise because $\boldsymbol{\Sigma}$ is diagonal and is real because each $\boldsymbol{\Sigma}_{ij} \geq 0$, which is true because $\mathbf{M}$ is positive semi-definite. Take $\mathbf{D} = \mathbf{R}$.

In our experiments we parameterize $\mathbf{D}$ as a diagonal-only matrix.

## C.4 INTEGRALS

The known integral requirement comes from the integrals required in the transition kernel, and can be relaxed two possible ways:

- numerical integration of function with unknown integral. This is expected to have low error given that the function is scalar-in scalar-out.
- Directly parameterize the integral and use auto-grad when needing the functions not-integrated.

We stick with the known integrals. In conclusion, the underlying parameters are positive scalar functions $b_q(s), b_d(s)$ defined on the time domain and with known integral, and general matrices $\tilde{\mathbf{Q}}_\phi, \tilde{\mathbf{D}}_\phi$.

## C.5 INSTANCES

**VPSDE.** VPSDE has $K = 1$. Consequently, $\mathbf{Q}, \mathbf{D}$ are $K \times K$. The only $1 \times 1$ skew-symmetric matrix is 0, so $\mathbf{Q} = 0$. Setting $\mathbf{D}(t) = \frac{1}{2}\beta(t)$ recovers VPSDE:

$$d\mathbf{y} = -\frac{\beta(t)}{2}\mathbf{y}dt + \sqrt{\beta(t)}d\mathbf{B}_t \tag{30}$$

$\nabla H(\mathbf{y}) = \mathbf{y}$ so $H(\mathbf{y}) = \frac{1}{2}\|\mathbf{y}\|_2^2$. The stationary distribution is $\mathcal{N}(0, \mathbf{I})$.

**CLD.** The CLD process (eq 5 in Dockhorn et al. (2021)) is defined as

$$\begin{pmatrix} d\mathbf{z}_t \\ d\mathbf{v}_r \end{pmatrix} = d\mathbf{y}_t = \begin{pmatrix} 0 & \frac{\beta}{M} \\ -\beta & -\frac{\Gamma\beta}{M} \end{pmatrix} \mathbf{y}_t + \begin{pmatrix} 0 & 0 \\ 0 & \sqrt{2\Gamma\beta} \end{pmatrix} d\mathbf{B}_t.$$

In $\mathbf{Q}/\mathbf{D}$ parameterization, we have

$$H(\mathbf{y}) = \frac{1}{2}\|\mathbf{z}\|_2^2 + \frac{1}{2M}\|\mathbf{v}\|_2^2, \qquad \nabla_\mathbf{u} H(\mathbf{y}) = \begin{pmatrix} \mathbf{z} \\ \frac{1}{M}\mathbf{v} \end{pmatrix}$$

$$\mathbf{Q} = \begin{pmatrix} 0 & -\beta \\ \beta & 0 \end{pmatrix}, \qquad \mathbf{D} = \begin{pmatrix} 0 & 0 \\ 0 & \Gamma\beta \end{pmatrix}$$

The stationary distribution of this process is:

$$q_{\phi,\infty} \propto \exp(-H(\mathbf{y})) = \mathcal{N}(\mathbf{z}; 0, I_d)\mathcal{N}(\mathbf{v}; 0, MI_d) \tag{31}$$

**ALDA.** Mou et al. (2019) define a third-order diffusion process for the purpose of gradient-based MCMC sampling. The ALDA diffusion process can be specified as

$$\mathbf{Q} = \begin{pmatrix} 0 & -\frac{1}{L}I & 0 \\ \frac{1}{L}I & 0 & -\gamma I \\ 0 & \gamma I & 0 \end{pmatrix}, \quad \mathbf{D} = \begin{pmatrix} 0 & 0 & 0 \\ 0 & 0 & 0 \\ 0 & 0 & \frac{\xi}{L}I \end{pmatrix}. \tag{32}$$

Note that $\mathbf{Q}$ is skew-symmetric and $\mathbf{D}$ is positive semi-definite, therefore we have that $q_t(\mathbf{u}) \rightarrow q_{\phi,\infty}$. In this case,

$$q_{\phi,\infty} = \mathcal{N}(\mathbf{z}; 0, \mathbf{I}_d)\mathcal{N}(\mathbf{v}_1; 0, \frac{1}{L}\mathbf{I}_d)\mathcal{N}(\mathbf{v}_2; 0, \frac{1}{L}\mathbf{I}_d)$$

MALDA. Similar to ALDA, we specify a diffusion process we term MALDA which we specify as

$$\mathbf{Q} = \begin{pmatrix} 0 & -\frac{1}{L}I & -\frac{1}{L} \\ \frac{1}{L}I & 0 & -\gamma I \\ \frac{1}{L} & \gamma I & 0 \end{pmatrix}, \quad \mathbf{D} = \begin{pmatrix} 0 & 0 & 0 \\ 0 & \frac{1}{L}I & 0 \\ 0 & 0 & \frac{1}{L}I \end{pmatrix}. \tag{33}$$

Note that $\mathbf{Q}$ is skew-symmetric and $\mathbf{D}$ is positive semi-definite. In this case this is

$$q_{\phi,\infty} = \mathcal{N}(\mathbf{z}; 0, \mathbf{I}_d)\mathcal{N}(\mathbf{v}_1; 0, \frac{1}{L}I_d)\mathcal{N}(\mathbf{v}_2; 0, \frac{1}{L}I_d)$$

## D    TRANSITIONS FOR LINEAR PROCESSES

For time variable $s$ and Brownian motion $\widehat{\mathbf{B}}_s$ driving diffusions of the form

$$d\mathbf{y} = f(\mathbf{y}, s)ds + g(s)d\widehat{\mathbf{B}}_s, \tag{34}$$

when $f_\phi(\mathbf{y}_s, s), g_\phi(s)$ are linear, the transition kernel $q_\phi(\mathbf{y}_s|\mathbf{y}_0)$ is always normal (Särkkä & Solin, 2019). Therefore, we just find the mean $\mathbf{m}_{s|0}$ and covariance $\boldsymbol{\Sigma}_{s|0}$ of $q(\mathbf{y}_s|\mathbf{y}_0)$. Let $f(\mathbf{y}, s) = \mathbf{A}(s)\mathbf{y}$. The un-conditional time $s$ mean and covariance are solutions to

$$d\mathbf{m}_s/ds = \mathbf{A}(s)\mathbf{m}_s$$
$$d\boldsymbol{\Sigma}_s/ds = \mathbf{A}(s)\boldsymbol{\Sigma}_s + \boldsymbol{\Sigma}_s\mathbf{A}^\top(s) + g^2(s) \tag{35}$$

By (6.6) in Särkkä & Solin (2019), for computing conditionals $q(\mathbf{y}_s|\mathbf{y}_0)$, we can take the marginal distribution ODEs and compute conditionals by simply setting the time 0 mean and covariance initial conditions to the conditioning value and to $\mathbf{0}$ respectively. We take (6.36-6.39) and set $\mathbf{m}_0 = \mathbf{u}_0$ and $\boldsymbol{\Sigma}_0 = 0$ to condition. Let $[\mathbf{A}]_s = \int_0^s \mathbf{A}(\nu)d\nu$. The mean is

$$\mathbf{m}_{s|0} = \exp\left[\int_0^s \mathbf{A}(\nu)d\nu\right]\mathbf{y}_0 = \exp\left(\left[A\right]_s\right) \underbrace{= \exp(s\mathbf{A})\mathbf{y}_0}_{\text{no integration if } \mathbf{A}(\nu) = \mathbf{A}}, \tag{36}$$

where $\exp$ denotes matrix exponential. (6.36-6.39) state the covariance $q(\mathbf{y}_s|\mathbf{y}_0)$ as a matrix factorization, for which a derivation is provided below $\boldsymbol{\Sigma}_s = \mathbf{C}_s(\mathbf{H}_s)^{-1}$ for $\mathbf{C}_s, \mathbf{H}_s$ being the solutions of:

$$\begin{pmatrix} \frac{d}{ds}\mathbf{C}_s \\ \frac{d}{ds}\mathbf{H}_s \end{pmatrix} = \begin{pmatrix} \mathbf{A}(s) & g^2(s) \\ \mathbf{0} & -\mathbf{A}^\top(s) \end{pmatrix} \begin{pmatrix} \mathbf{C}_s \\ \mathbf{H}_s \end{pmatrix} \tag{37}$$

To condition and get $\boldsymbol{\Sigma}_{s|0}$ from $\boldsymbol{\Sigma}_s$, we set $\boldsymbol{\Sigma}_0 = \mathbf{0}$, and initialize $\mathbf{C}_s, \mathbf{H}_s$ by $\mathbf{C}_0 = \mathbf{0}$ and $\mathbf{H}_0 = \mathbf{I}$.

$$\begin{pmatrix} \mathbf{C}_s \\ \mathbf{H}_s \end{pmatrix} = \exp\left[\begin{pmatrix} [\mathbf{A}]_s & [g^2]_s \\ \mathbf{0} & -[\mathbf{A}^\top]_s \end{pmatrix}\right]\begin{pmatrix} \mathbf{0} \\ \mathbf{I} \end{pmatrix} \underbrace{= \exp\left[s\begin{pmatrix} \mathbf{A} & g^2 \\ \mathbf{0} & -\mathbf{A}^\top \end{pmatrix}\right]\begin{pmatrix} \mathbf{0} \\ \mathbf{I} \end{pmatrix}}_{\text{no integration if } \mathbf{A}(\nu) = \mathbf{A}, g(\nu) = g}. \tag{38}$$

Finally, $\boldsymbol{\Sigma}_{s|0} = \mathbf{C}_s(\mathbf{H}_s)^{-1}$.

### D.1    DERIVATION OF THE COVARIANCE MATRIX SOLUTION

Equation (35) gives an expression for $d\boldsymbol{\Sigma}_s/ds$. To derive the matrix factorization technique used in eq. (37), we use eq. (35) and the desired condition $\boldsymbol{\Sigma}_s = \mathbf{C}_s\mathbf{H}_s^{-1}$ to derive expressions for $d\mathbf{C}_s/ds$ and $d\mathbf{H}_s/ds$ and suitable intial conditions so that the factorization also starts at the desired $\boldsymbol{\Sigma}_0$. Let $\boldsymbol{\Sigma}_s = \mathbf{C}_s\mathbf{H}_s^{-1}$, then note that $\mathbf{C}_s, \mathbf{H}_s$ satisfies

$$\frac{d}{ds}\boldsymbol{\Sigma}_s = \frac{d}{ds}\mathbf{C}_s\mathbf{H}_s^{-1}$$
$$= \mathbf{C}_s\frac{d}{ds}\mathbf{H}_s^{-1} + \left(\frac{d}{ds}\mathbf{C}_s\right)\mathbf{H}_s^{-1}$$

And using the fact that

$$\frac{d}{ds}\mathbf{H}_s\mathbf{H}_s^{-1} = 0$$

$$\mathbf{H}_s\frac{d}{ds}\mathbf{H}_s^{-1} + \frac{d}{ds}\mathbf{H}_s\left(\mathbf{H}_s^{-1}\right) = 0$$

$$\frac{d}{ds}\mathbf{H}_s^{-1} = -\mathbf{H}_s^{-1}\frac{d}{ds}\mathbf{H}_s\left(\mathbf{H}_s^{-1}\right)$$

we get that

$$\mathbf{C}_s\frac{d}{ds}\mathbf{H}_s^{-1} + \left(\frac{d}{ds}\mathbf{C}_s\right)\mathbf{H}_s^{-1} = -\mathbf{C}_s\mathbf{H}_s^{-1}\frac{d}{ds}\mathbf{H}_s\left(\mathbf{H}_s^{-1}\right) + \left(\frac{d}{ds}\mathbf{C}_s\right)\mathbf{H}_s^{-1}$$

$$-\mathbf{C}_s\mathbf{H}_s^{-1}\frac{d}{ds}\mathbf{H}_s\left(\mathbf{H}_s^{-1}\right) + \left(\frac{d}{ds}\mathbf{C}_s\right)\mathbf{H}_s^{-1} = \mathbf{A}(s)\mathbf{C}_s\mathbf{H}_s^{-1} + \mathbf{C}_s\mathbf{H}_s^{-1}\mathbf{A}^\top(s) + g^2(s)$$

$$= \mathbf{A}(s)\mathbf{C}_s\mathbf{H}_s^{-1} + \mathbf{C}_s\mathbf{H}_s^{-1}\mathbf{A}^\top(s)\mathbf{H}_s\mathbf{H}_s^{-1} + g^2(s)\mathbf{H}_s\mathbf{H}_s^{-1}$$

$$\left(-\mathbf{C}_s\mathbf{H}_s^{-1}\frac{d}{ds}\mathbf{H}_s + \frac{d}{ds}\mathbf{C}_s\right)\mathbf{H}_s^{-1} = \left(\mathbf{A}(s)\mathbf{C}_s + \mathbf{C}_s\mathbf{H}_s^{-1}\mathbf{A}^\top(s)\mathbf{H}_s + g^2(s)\mathbf{H}_s\right)\mathbf{H}_s^{-1}$$

$$-\mathbf{C}_s\mathbf{H}_s^{-1}\frac{d}{ds}\mathbf{H}_s + \frac{d}{ds}\mathbf{C}_s = \mathbf{A}(s)\mathbf{C}_s + \mathbf{C}_s\mathbf{H}_s^{-1}\mathbf{A}^\top(s)\mathbf{H}_s + g^2(s)\mathbf{H}_s$$

$$\begin{bmatrix}\mathbf{C}_s\mathbf{H}_s^{-1} & \mathbf{I}_d\end{bmatrix}^\top \frac{d}{ds}\begin{pmatrix}\mathbf{H}_s \\ \mathbf{C}_s\end{pmatrix} = \begin{bmatrix}\mathbf{C}_s\mathbf{H}_s^{-1} & \mathbf{I}_d\end{bmatrix}^\top \begin{pmatrix}-\mathbf{A}^\top(s)\mathbf{H}_s \\ \mathbf{A}(s)\mathbf{C}_s + g^2(s)\mathbf{H}_s\end{pmatrix}$$

Now, we note $\mathbf{C}_s, \mathbf{H}_s$ satisfy the following

$$\frac{d}{ds}\mathbf{H}_s = -\mathbf{A}^\top(s)\mathbf{H}_s$$

$$\frac{d}{ds}\mathbf{C}_s = \mathbf{A}(s)\mathbf{C}_s + g^2(s)\mathbf{H}_s$$

which implies that

$$\frac{d}{ds}\begin{pmatrix}\mathbf{C}_s \\ \mathbf{H}_s\end{pmatrix} = \begin{pmatrix}\mathbf{A}(s) & g^2(s) \\ \mathbf{0} & -\mathbf{A}^\top(s)\end{pmatrix}\begin{pmatrix}\mathbf{C}_s \\ \mathbf{H}_s\end{pmatrix} \tag{39}$$

with $\mathbf{C}_0 = \mathbf{\Sigma}_0$ and $\mathbf{H}_0 = \mathbf{I}_d$, as $\mathbf{C}_0\mathbf{H}_0^{-1} = \mathbf{\Sigma}_0$.

### D.2 HYBRID SCORE MATCHING

Instead of computing $q(\mathbf{y}_s|\mathbf{y}_0)$, we can apply the hybrid score matching principle (Dockhorn et al., 2021) to reduce variance by compute objectives using $q(\mathbf{y}_s|x)$ instead of $q(\mathbf{y}_s|\mathbf{y}_0)$, which amounts to integrating out $\mathbf{v}_0$. To accomplish this, following Särkkä & Solin (2019), we simply replace $\mathbf{y}_0$ with $[x, \mathbb{E}[\mathbf{v}_0]]$ in the expression for $\mathbf{m}_{s|0}$, i.e. replace the conditioning value of $\mathbf{v}_0$ with the mean of its chosen initial distribution:

$$\mathbb{E}[\mathbf{y}_s|x] = \exp\left[\int_0^s A(\nu)d\nu\right]\begin{pmatrix}x \\ \mathbb{E}[\mathbf{v}_0]\end{pmatrix} \tag{40}$$

For the convariance, instead of using $\mathbf{C}_0 = \mathbf{\Sigma}_0 = \mathbf{0}$, we use a block matrix to condition on $x$ but not $\mathbf{v}_0$. We decompose $\mathbf{\Sigma}_0$ into its blocks $\mathbf{\Sigma}_{0,xx}$, $\mathbf{\Sigma}_{0,vv}$ ,$\mathbf{\Sigma}_{0,xv}$. As before, to condition on $x$ we set $\mathbf{\Sigma}_{0,xx} = \mathbf{0}$. Because $q(\mathbf{v}_0)$ is set to be independent of $x$, $\mathbf{\Sigma}_{0,xv}$ is also set to $\mathbf{0}$. Finally, instead of $\mathbf{0}$, to marginalize out $\mathbf{v}_0$, $\mathbf{\Sigma}_{0,vv}$ is set to the covariance of the chosen initial time zero distribution for $\mathbf{v}_0$. E.g. if $\mathbf{v}_{0,j} \sim N(0,\gamma)$ for each dimension, then $\mathbf{\Sigma}_{0,vv} = N(0,\gamma I)$.

We operationalize this in a simple piece of code, which makes the ELBO tractable and easy, i.e. skips both analytic derivations and numerical forward integration during training.

### D.3 Transitions in Stationary Parameterization

In terms of $\mathbf{Q}, \mathbf{D}$, the transitions $q(\mathbf{y}_s | \mathbf{y}_0)$ for time $s$ are normal with mean $\mathbf{m}_{s|0}$ and $\mathbf{\Sigma}_{s|0}$ equal to:

$$\mathbf{m}_{s|0} = \exp\left(-\left[\mathbf{Q}+\mathbf{D}\right]_s\right)\mathbf{y}_0, \qquad \begin{pmatrix} \mathbf{C}_s \\ \mathbf{H}_s \end{pmatrix} = \exp\left[\begin{pmatrix} -[\mathbf{Q}+\mathbf{D}]_s & [2\mathbf{D}]_s \\ \mathbf{0} & [(\mathbf{Q}+\mathbf{D})^\top]_s \end{pmatrix}\right]\begin{pmatrix} \mathbf{0} \\ \mathbf{I} \end{pmatrix} \tag{41}$$

where $\mathbf{\Sigma}_{s|0} = \mathbf{C}_s(\mathbf{H}_s)^{-1}$. For the time invariant case, this simplifies to

$$\mathbf{m}_{s|0} = \exp[-s(\mathbf{Q}+\mathbf{D})]\mathbf{y}_0, \qquad \begin{pmatrix} \mathbf{C}_s \\ \mathbf{H}_s \end{pmatrix} = \exp\left[s\begin{pmatrix} -(\mathbf{Q}+\mathbf{D}) & 2\mathbf{D} \\ \mathbf{0} & (\mathbf{Q}+\mathbf{D})^\top \end{pmatrix}\right]\begin{pmatrix} \mathbf{0} \\ \mathbf{I} \end{pmatrix} \tag{42}$$

## E  Generic change of measure and Jensen's for approximate marginalization

Suppose $\mathbf{u} = [\mathbf{z}, \mathbf{v}]$ and we have an expression for $p(\mathbf{u} = [z, v]) = p(\mathbf{z} = z, \mathbf{v} = v)$. By marginalization, we can get $p(\mathbf{z} = z)$, and we can introduce another distribution $q$ to pick a sampling distribution of our choice:

$$
\begin{aligned}
p(\mathbf{z} = z) &= \int_v p(\mathbf{z} = z, \mathbf{v} = v)dv \\
&= \int_v p(\mathbf{z} = z | \mathbf{v} = v)p(\mathbf{v} = v)dv \\
&= \int_v \frac{q(\mathbf{v} = v | \mathbf{z} = z)}{q(\mathbf{v} = v | \mathbf{z} = z)}p(\mathbf{z} = z | \mathbf{v} = v)p(\mathbf{v} = v)dv \\
&= \mathbb{E}_{q(\mathbf{v}=v|\mathbf{z}=z)}\left[\frac{p(\mathbf{z} = z, \mathbf{v} = v)}{q(\mathbf{v} = v | \mathbf{z} = z)}\right]
\end{aligned}
\tag{43}
$$

We often work with these expressions in log space, and need to pull the expectation outside to use Monte Carlo. Jensen's bound allows this:

$$
\begin{aligned}
\log p(\mathbf{z} = z) &= \log \mathbb{E}_{q(\mathbf{v}=v|\mathbf{z}=z)}\left[\frac{p(\mathbf{z} = z, \mathbf{v} = v)}{q(\mathbf{v} = v | \mathbf{z} = z)}\right] \\
&\geq \mathbb{E}_{q(\mathbf{v}=v|\mathbf{z}=z)}\left[\log \frac{p(\mathbf{z} = z, \mathbf{v} = v)}{q(\mathbf{v} = v | \mathbf{z} = z)}\right]
\end{aligned}
$$

The following shows that the bound is tight when $q(\mathbf{v} = v | \mathbf{z} = z) = p(\mathbf{v} = v | \mathbf{z} = z)$:

$$
\begin{aligned}
\mathbb{E}_{q(\mathbf{v}=v|\mathbf{z}=z)}\left[\log \frac{p(\mathbf{z} = z, \mathbf{v} = v)}{q(\mathbf{v} = v | \mathbf{z} = z)}\right] &=_{\text{assume}} \mathbb{E}_{p(\mathbf{v}=v|\mathbf{z}=z)}\left[\log \frac{p(\mathbf{z} = z, \mathbf{v} = v)}{p(\mathbf{v} = v | \mathbf{z} = z)}\right] \\
&= \mathbb{E}_{p(\mathbf{v}=v|\mathbf{z}=z)}\left[\log\left(\frac{p(\mathbf{z} = z, \mathbf{v} = v)}{p(\mathbf{v} = v, \mathbf{z} = z)} \cdot p(\mathbf{z} = z)\right)\right] \\
&= \mathbb{E}_{p(\mathbf{v}=v|\mathbf{z}=z)}\left[\log p(\mathbf{z} = z)\right] \\
&= \log p(\mathbf{z} = z)
\end{aligned}
\tag{44}
$$

# F  ELBO FOR MDMs

$$\log p_\theta(x) = \log \int_{v_0} p_\theta(x_0, v_0) dv_0 \tag{45}$$

$$= \log \int_{v_0} p_\theta(u_0 = [x, v_0]) \tag{46}$$

$$= \log \int_{v_0} \frac{q(v_0|x)}{q(v_0|x)} p_\theta(u_0 = [x, v_0]) \tag{47}$$

$$= \log \mathbb{E}_{q(v_0|x)} \left[ \frac{p_\theta(u_0 = [x, v_0])}{q(v_0|x)} \right] \tag{48}$$

$$\geq \mathbb{E}_{q(v_0|x)} \left[ \log p_\theta(u_0 = [x, v_0]) - \log q(v_0|x) \right] \tag{49}$$

$$\geq \mathbb{E}_{q(y|x)} \left[ \log \pi_\theta(y_T) + \int_0^T -\|s_\theta\|_{g^2}^2 - \nabla \cdot (g^2 s_\theta - f) ds - \log q(y_0^v|x) \right] \tag{50}$$

The first inequality holds due to Jensen's inequality and the second due to an application of Theorem 1 from Huang et al. (2021) or Theorem 3 from Song et al. (2021) applied to the joint variable $\mathbf{u}_0$.

## F.1  ISM TO DSM

### F.1.1  LEMMA: EXPECTATION BY PARTS

We will need a form of multivariate integration by parts which gives us for some $f$ and some $q(x)$,
$E_{q(x)}[\nabla_x \cdot f(x)] = -E_{q(x)}[f(x)^\top \nabla_x \log q(x)]$

$$E_{q(x)}[\nabla_x \cdot f_i(x)] = \int q(x) \sum_{i=1}^d [\nabla_{x_i} f_i(x)] dx$$

$$= \int \sum_{i=1}^d q(x) \nabla_{x_i} f_i(x) dx$$

$$= \sum_{i=1}^d \int_{x_{-i}} \int_{x_i} q(x) \nabla_{x_i} f_i(x) dx_i dx_{-i}$$

$$= \sum_{i=1}^d \int \left[ \left[ q(x) \int \nabla_{x_i} f_i(x) dx_i \right]_{-\infty}^{\infty} - \int \nabla_{x_i} q(x) \int \nabla_{x_i} f_i(x) dx_i \right] dx_{-i}$$

$$= \sum_{i=1}^d \int \left[ -\int \nabla_{x_i} q(x) f_i(x) dx_i \right] dx_{-i}$$

$$= \sum_{i=1}^d \int \left[ -\int q(x) \nabla_{x_i} \log q(x) f_i(x) dx_i \right] dx_{-i}$$

$$= \sum_{i=1}^d -\int \int q(x) \nabla_{x_i} \log q(x) f_i(x) dx_i dx_{-i}$$

$$= \sum_{i=1}^d -E_{q(x)} \left[ \nabla_{x_i} \log q(x) f_i(x) \right]$$

$$= -E_{q(x)}[f(x)^\top \nabla_x \log q(x)]$$

This equality also follows directly from the Stein operator using the generator method to the Langevin diffusion (Barbour, 1988).

### F.1.2  DSM ELBO

Using the "expectation by parts", we have:

$$\mathbb{E}_{q(u_t|x)}[\nabla_{u_t} \cdot g^2(t)s_\theta(u_t,t)] = -\mathbb{E}_{q(u_t|x)}[(g^2(t)s_\theta(u_t,t))^\top \nabla_{u_t} \log q(u_t|x)]$$

Also we have, for $s_\theta$ evaluated at $(u_t,t)$, by completing the square,

$$-\frac{1}{2}||s_\theta||_{g^2(t)} + s_\theta^\top g^2(t)\nabla \log q(u_t|x) = -\frac{1}{2}||s_\theta - \nabla \log q(u_t|x)||^2_{g^2(t)} + .5||\nabla \log q(u_t|x)||^2_{g^2(t)}$$

The two together give us:

$$\log p(x) \geq \mathbb{E}_{q(u_T|x)}\left[\log \pi\right] + \int_0^T \left[\mathbb{E}_{q(u_t|x)}\left[-\nabla \cdot g^2 s_\theta - .5||s_\theta||^2_{g^2(t)} + \nabla \cdot f\right]dt\right]$$

$$= \mathbb{E}_{q(u_T|x)}\left[\log \pi\right] + \int_0^T \left[\mathbb{E}_{q(u_t|x)}\left[(g^2 s_\theta)^\top \nabla_{u_t} \log q(u_t|x) - .5||s_\theta||^2_{g^2(t)} + \nabla \cdot f\right]dt\right]$$

$$= \mathbb{E}_{q(u_T|x)}\left[\log \pi\right] + \int_0^T \left[\mathbb{E}_{q(u_t|x)}\left[-\frac{1}{2}||s_\theta - \nabla \log q(u_t|x)||^2_{g^2(t)}\right.\right.$$

$$\left.\left. + .5||\nabla \log q(u_t|x)||^2_{g^2(t)} + \nabla_{u_t} \cdot f\right]\right]dt$$

$$(51)$$

### F.2  NOISE PREDICTION

We have that for normal $\mathcal{N}(\mathbf{y}_s; \mathbf{m}_{s|0}, \mathbf{\Sigma}_{s|0})$, we can sample $\mathbf{y}_s$ with normal noise $\epsilon \sim \mathcal{N}(0, I)$ and $\mathbf{y}_s = \mathbf{m}_{s|0} + \mathbf{L}\epsilon$ where $\mathbf{L}$ is the cholesky decomposition of $\mathbf{\Sigma}_{s|0}$ Then, the score is

$$\nabla_{\mathbf{y}_s} \log q(\mathbf{y}_s|\mathbf{y}_0)\Big|_{\mathbf{y}_s=\mathbf{m}_{s|0}+\mathbf{L}\epsilon}$$

$$= -\mathbf{\Sigma}_{s|0}^{-1}\left(\mathbf{y}_s - \mathbf{m}_{s|0}\right)$$

$$= -\mathbf{\Sigma}_{s|0}^{-1}\left(\left[\mathbf{m}_{s|0} + \mathbf{L}\epsilon\right] - \mathbf{m}_{s|0}\right)$$

$$= -\mathbf{\Sigma}_{s|0}^{-1}\left(\mathbf{L}\epsilon\right)$$

$$= -\left(\mathbf{L}\mathbf{L}^\top\right)^{-1}\left(\mathbf{L}\epsilon\right)$$

$$= -\left(\mathbf{L}^\top\right)^{-1}\mathbf{L}^{-1}\mathbf{L}\epsilon$$

$$= -\left(\mathbf{L}^\top\right)^{-1}\epsilon = -\left(\mathbf{L}^{-1}\right)^\top \epsilon = -\mathbf{L}^{\top,-1}\epsilon$$

Parameterize $s_\theta(\mathbf{y}_s, s)$ as $s_\theta(\mathbf{y}_s, s) = -\mathbf{L}^{\top,-1}\epsilon_\theta(\mathbf{y}, s)$. This gives

$$\frac{1}{2}||-\mathbf{L}^{\top,-1}\epsilon_\theta(\mathbf{y}, s) \quad - \quad -\mathbf{L}^{\top,-1}\epsilon||^2_{g^2_\phi(s)}$$

$$= \frac{1}{2}||\mathbf{L}^{\top,-1}\epsilon \quad - \quad \mathbf{L}^{\top,-1}\epsilon_\theta(\mathbf{y}, s)||^2_{g^2_\phi(s)}$$

$$= \frac{1}{2}\left(\mathbf{L}^{\top,-1}\epsilon \quad - \quad \mathbf{L}^{\top,-1}\epsilon_\theta(\mathbf{y}, s)\right)^\top g^2_\phi(s)\left(\mathbf{L}^{\top,-1}\epsilon \quad - \quad \mathbf{L}^{\top,-1}\epsilon_\theta(\mathbf{y}, s)\right)$$

$$= \frac{1}{2}\left(\mathbf{L}^{\top,-1}\left[\epsilon - \epsilon_\theta(\mathbf{y}, s)\right]\right)^\top g^2_\phi(s)\left(\mathbf{L}^{\top,-1}\left[\epsilon - \epsilon_\theta(\mathbf{y}, s)\right]\right)$$

We can also use this insight to analytically compute the quadratic score term (following is computed per data-dimension, so must be multiplied by $D$ when computing the ELBO):

$$
\mathbb{E}_{\mathbf{y}_0}\mathbb{E}_{\mathbf{y}_s|\mathbf{y}_0}\left[\frac{1}{2}\|\nabla_{\mathbf{y}_s}\log q_\phi(\mathbf{y}_s|\mathbf{y}_0)\|^2_{g^2_\phi(s)}\right] = \mathbb{E}_{\mathbf{y}_0}\mathbb{E}_{\mathbf{y}_s|\mathbf{y}_0}\left[\left(\nabla_{\mathbf{y}_s}\log q_\phi(\mathbf{y}_s|\mathbf{y}_0)\right)^\top g^2_\phi(s)\left(\nabla_{\mathbf{y}_s}\log q_\phi(\mathbf{y}_s|\mathbf{y}_0)\right)\right]
$$

$$
= \mathbb{E}_{\mathbf{y}_0}\mathbb{E}_{\mathbf{y}_s|\mathbf{y}_0}\left[\left(-\mathbf{L}^{\top,-1}\epsilon\right)^\top g^2_\phi(s)\left(-\mathbf{L}^{\top,-1}\epsilon\right)\right]
$$

$$
= \mathbb{E}_{\mathbf{y}_0}\mathbb{E}_{\mathbf{y}_s|\mathbf{y}_0}\left[\epsilon^\top(-\mathbf{L}^{-1})g^2_\phi(s)(-\mathbf{L}^{\top,-1})\epsilon\right]
$$

$$
= \mathbb{E}_{\mathbf{y}_0}\mathbb{E}_{\mathbf{y}_s|\mathbf{y}_0}\left[\epsilon^\top\left(\mathbf{L}^{-1}g^2_\phi(s)\mathbf{L}^{\top,-1}\right)\epsilon\right]
$$

$$
= \mathbb{E}_{\mathbf{y}_0}\mathbb{E}_\epsilon\left[\epsilon^\top\left(\mathbf{L}^{-1}g^2_\phi(s)\mathbf{L}^{\top,-1}\right)\epsilon\right]
$$

$$
= \mathbb{E}_\epsilon\left[\epsilon^\top\left(\mathbf{L}^{-1}g^2_\phi(s)\mathbf{L}^{\top,-1}\right)\epsilon\right]
$$

$$
= \text{Trace}\left(\mathbf{L}^{-1}g^2_\phi(s)\mathbf{L}^{\top,-1}\right)
$$

## G  ELBOS IN STATIONARY PARAMETERIZATION

We use the stationary parmeterization described in appendix C. We now specialize the ELBO to the linear stationary parameterization.

Recall $f_\phi(\mathbf{y}, s) = -[\mathbf{Q}_\phi(s) + \mathbf{D}_\phi(s)]\mathbf{y}$. Recall $g_\phi(s) = \sqrt{2\mathbf{D}_\phi(s)}$ We have $g^2_\phi(s) = 2\mathbf{D}_\phi(s)$. We can write the MDM ISM ELBO as

$$
\mathcal{L}^{\text{mism}} = \mathbb{E}_{v\sim q_\gamma}\left[\mathbb{E}_{s\sim\text{Unif}(0,T)}\left[\ell^{(ism)}_s\right] + \ell_T + \ell_q\right] \tag{52}
$$

where

$$
\ell_{s_\theta} = -\frac{1}{2}\|s_\theta(\mathbf{y}_s, s)\|^2_{\underbrace{2\mathbf{D}_\phi(s)}_{g^2_\phi}}
$$

$$
\ell_{\text{div-fgs}} = \nabla_{\mathbf{y}_s}\cdot\left[\underbrace{-[\mathbf{Q}_\phi(s) + \mathbf{D}_\phi(s)]\mathbf{y}_s}_{f_\phi} - \underbrace{2\mathbf{D}_\phi(s)}_{g^2_\phi}s_\theta(\mathbf{y}_s, s)\right]
$$

$$
\ell^{\text{ism}}_s = \mathbb{E}_{\underbrace{q_{\phi,s,(x,v)}}_{\text{depends on }\mathbf{Q},\mathbf{D}}}\left[\ell_{s_\theta} + \ell_{\text{div-fgs}}\right] \tag{53}
$$

$$
\ell_T = \mathbb{E}_{\underbrace{q_{\phi,T},(x,v)}_{\text{depends on }\mathbf{Q},\mathbf{D}}}\left[\log\pi_\theta(\mathbf{y}_T)\right]
$$

$$
\ell_q = -\log q_\gamma(v|x)
$$

For the DSM form,

$$
\mathcal{L}^{\text{mdsm}} = \mathbb{E}_{v\sim q_\gamma}\left[\mathbb{E}_{s\sim\text{Unif}(0,T)}\left[\ell^{(dsm)}_s\right] + \ell_T + \ell_q\right] \tag{54}
$$

where

$$\ell_{\text{div-f}} = \nabla_{\mathbf{y}_s} \cdot \underbrace{-[\mathbf{Q}_\phi(s) + \mathbf{D}_\phi(s)]\mathbf{y}_s}_{f_\phi}$$

$$\ell_{\text{fwd-score}} = \frac{1}{2} \Big|\Big| \underbrace{\nabla_{\mathbf{y}_s} \log q_\phi(\mathbf{y}_s|\mathbf{y}_0)}_{\text{depends on } \mathbf{Q},\mathbf{D}} \Big|\Big|^2_{2\underbrace{\mathbf{D}_\phi(s)}_{g_\phi^2}}$$

$$\ell_{\text{neg-scorediff}} = -\frac{1}{2} \|s_\theta(\mathbf{y}_s, s) - \underbrace{\nabla_{\mathbf{y}_s} \log q_\phi(\mathbf{y}_s|\mathbf{y}_0)}_{\text{depends on } \mathbf{Q},\mathbf{D}} \|^2_{2\underbrace{\mathbf{D}_\phi(s)}_{g_\phi^2}}$$

$$\ell_s^{(dsm)} = \mathbb{E}_{\underbrace{q_{\phi,s,(x,v)}}_{\text{depends on } \mathbf{Q},\mathbf{D}}} \left[ \ell_{\text{neg-scorediff}} + \ell_{\text{fwd-score}} + \ell_{\text{div-f}} \right]$$

# H  ALGORITHMS

## H.1  GENERIC TRANSITION KERNEL

---

**Algorithm 2** Get transition distribution $\mathbf{y}_s|x$

---

**Input:** data $x$. time $s$. $\mathbf{A}, g$.
**compute:** $\mathbf{A}(s)$ and $g(s)$
**compute:** $\mathbf{M}_s = \int_0^s \mathbf{A}(t)dt$ (integrated drift)
**compute:** $\mathbf{N}_s = \int_0^s g^2(t)dt$ (integrated diffusions squared)
**compute:** $\gamma_{s|0} = \exp\left(\mathbf{M}_s\right)$ (mean coefficient)
**set:** $\mathbf{y}_0 = [x, 0_1, \ldots, 0_{K-1}]$, $\mathbf{\Sigma}_{0,zz} = \mathbf{0}$, and $\mathbf{\Sigma}_{0,zv}, \mathbf{\Sigma}_{0,vv}$ to chosen initial distribution
**compute:** $\mathbf{m}_{s|0} = \gamma_{s|0}\mathbf{y}_0$ (**mean**)
**compute:**

$$\begin{pmatrix} \mathbf{C}_s \\ \mathbf{H}_s \end{pmatrix} = \exp\left[ \begin{pmatrix} \mathbf{M}_s & \mathbf{N}_s \\ \mathbf{0} & -\mathbf{M}_s^\top \end{pmatrix} \right] \begin{pmatrix} \mathbf{\Sigma}_0 \\ \mathbf{I} \end{pmatrix} \quad \text{(ingredients for cov.)} \tag{55}$$

**compute:** $\mathbf{\Sigma}_{s|0} = \mathbf{C}_s(\mathbf{H}_s)^{-1}$  (**cov.**)
**Output:** $\mathcal{N}(\mathbf{m}_{s|0}, \mathbf{\Sigma}_{s|0})$

---

## H.2  TRANSITIONS WITH $Q, D$

Current param matrices $\tilde{\mathbf{Q}}_\phi, \tilde{\mathbf{D}}_\phi$ and along with fixed time-in scalar-out functions $b_q(s), b_d(s)$ and their known integrals $B_q(s), B_d(s)$. $q_\gamma(v_0|z_0 = x)$ taken to be parameterless so that $v_0 \sim \mathcal{N}(0, I)$. Model params are $s_\theta$ fixed $\pi_\theta$.

---

**Algorithm 3** Get $\mathbf{Q}, \mathbf{D}$ and their integrated terms $\mathbf{M}, \mathbf{N}$

---

**Input:** time $s$ and current params $\phi$
**compute:** $[b_q]_s = \int_0^s b_q(\nu)d\nu$ using known integral $B_q(s) - B_q(0)$
**compute:** $[b_d]_s = \int_0^s b_d(\nu)d\nu$ using known integral $B_d(s) - B_d(0)$.
**compute:** $[\mathbf{Q}_\phi]_s = [b_q]_s \cdot \left[\tilde{\mathbf{Q}}_\phi - \tilde{\mathbf{Q}}_\phi^\top\right]$ for current params $\tilde{\mathbf{Q}}_\phi$.
**compute:** $[\mathbf{D}_\phi]_s = [b_d]_s \cdot \left[\tilde{\mathbf{D}}_\phi\tilde{\mathbf{D}}_\phi^\top\right]$ for current params $\tilde{\mathbf{D}}_\phi$.
**compute:** $\mathbf{M}_s = -([\mathbf{Q}_\phi]_s + [\mathbf{D}_\phi]_s)$ ($\mathbf{M}$ just a variable name)
**compute:** $\mathbf{N}_s = [2\mathbf{D}_\phi]_s = 2 \cdot [\mathbf{D}_\phi]_s$ ($\mathbf{N}$ just a variable name)
**compute:** $\mathbf{Q}_s = b_q(s) \cdot \left[\tilde{\mathbf{Q}}_\phi - \tilde{\mathbf{Q}}_\phi^\top\right]$ (not integrated)
**compute:** $\mathbf{D}_s = b_d(s) \cdot \left[\tilde{\mathbf{D}}_\phi\tilde{\mathbf{D}}_\phi^\top\right]$ (not integrated)
**compute:** $A_s = -[\mathbf{Q}_s + \mathbf{D}_s]$ (drift coef.)
**compute:** $g_s^2 = 2\mathbf{D}_s$ (diffusion coef. squared)
**Output:** $\mathbf{A}_s, g_s^2, \mathbf{M}_s, \mathbf{N}_s$

---

## H.3  ELBO ALGORITHMS

---

**Algorithm 4** Get transition distributions

---

**Input:** Sample $\mathbf{y}_0 = (x, v)$ and time $s$. Current params $\phi$
**set:** $\mathbf{A}_s, g_s^2, \mathbf{M}_s, \mathbf{N}_s \leftarrow$ algorithm 3
**compute:** $\mathbf{m}_{s|0} = \exp\left(\mathbf{M}_s\right)\mathbf{y}_0$ (transition mean)
**compute:** ingredients for transition cov. matrix:

$$\begin{pmatrix} \mathbf{C}_s \\ \mathbf{H}_s \end{pmatrix} = \exp\left[ \begin{pmatrix} \mathbf{M}_s & \mathbf{N}_s \\ \mathbf{0} & -\mathbf{M}_s^\top \end{pmatrix} \right] \begin{pmatrix} \mathbf{0} \\ \mathbf{I} \end{pmatrix} \tag{56}$$

**compute:** $\boldsymbol{\Sigma}_{s|0} = \mathbf{C}_s(\mathbf{H}_s)^{-1}$ (transition cov).
**instantiate:** $q_{\phi,s,(x,v)} = q_\phi(\mathbf{y}_s|\mathbf{y}_0) = \mathcal{N}(\mathbf{m}_{s|0}, \boldsymbol{\Sigma}_{s|0})$.
**Output:** $q_{\phi,s,(x,v)}, A_s, g_s^2$

---

**Algorithm 5** Compute ELBO with ism or dsm

---

**input:** Data point $x$ and current params $\theta, \phi, \gamma$
**draw:** an aux. sample $v \sim q_\gamma(v|x)$
**draw:** a sample $s \sim \text{Unif}(0, T)$
**set:** $\mathbf{y}_0 = (x, v)$
**set:** $q_{\phi,s,\mathbf{y}_0}, A_s, g_s^2 \leftarrow$ algorithm 4 called on $\mathbf{y}_0, s, \phi$
**draw:** $\mathbf{y}_s \sim q_{\phi,s,\mathbf{y}_0}$
**compute:** $\ell_s$ with dsm$(s)$ (algorithm 6) or ism$(s)$ (algorithm 7) on $\mathbf{y}_s, \theta, A_s, g_s^2, q_{\phi,s,\mathbf{y}_0}$
**set:** $q_{\phi,T,\mathbf{y}_0}, \text{--}, \text{--} \leftarrow$ algorithm 4 called on $\mathbf{y}_0, T, \phi$
**draw:** $\mathbf{y}_T \sim q_{\phi,T,\mathbf{y}_0}$
**output:** $\ell_s + \log \pi_\theta(\mathbf{y}_T) - \log q_\gamma(v)$

---

**Algorithm 6** Compute dsm$(s)$

---

**input:** $\mathbf{y}_s, \theta, A_s, g_s^2, q_{\phi,s,\mathbf{y}_0}$.
**compute:** fwd-score $= \nabla_{\mathbf{y}_s} \log q_\phi(\mathbf{y}_s|\mathbf{y}_0)$
**compute:** model-score $= s_\theta(\mathbf{y}_s, s)$
**compute:** fwd-score-term $= \frac{1}{2}(\text{fwd-score})^\top g_s^2(\text{fwd-score})$
**compute:** score-diff $=$ model-score $-$ fwd-score
**compute:** diff-term $= -\frac{1}{2}\text{score-diff}^\top g_s^2 \text{score-diff}$
**compute:** div-f $= \nabla_{\mathbf{y}_s} \cdot A_s\mathbf{y}_s$
**output:** dsm$(s) =$ fwd-score-term $+$ diff-term $+$ div-f

---

**Algorithm 7** Compute ism$(s)$

---

**input:** $\mathbf{y}_s, \theta, A_s, g_s^2, q_{\phi,s,\mathbf{y}_0}$.
**compute:** model-score $= s_\theta(\mathbf{y}_s, s)$
**compute:** score-term $= -\frac{1}{2}\text{model-score}^\top g_s^2 \text{model-score}$
**compute:** div-gs $= \nabla_{\mathbf{y}_s} \cdot g_s^2 s_\theta(\mathbf{y}_s, s)$
**compute:** div-f $= \nabla_{\mathbf{y}_s} \cdot A_s\mathbf{y}_s$
**compute:** div-term $= -\text{div-gs} + \text{div-f}$
**output:** ism$(s) =$ score-term $+$ div-term

---

# I  VALID ELBO WITH TRUNCATION

The integrand in the ELBO and its gradients is not bounded at time 0. Therefore, following Sohl-Dickstein et al. (2015) and Song et al. (2021) the integrand in eq. (7) is integrated from $[\epsilon, T]$, rather than $[0, T]$. However, that integral is not a valid lower bound on $\log p_\theta(x)$. Instead, it can be viewed as a proper lower bound on the prior for a latent variable $\mathbf{y}_\epsilon$. Therefore, to provide a bound for the data, one can introduce a likelihood and substitute the prior lower bound into a standard variational bound that integrates out the latent.

To provide a valid lower bound for multivariate diffusions, we extend theorem 6 in Song et al. (2021) from univariate to multivariate diffusions.

**Theorem 3.** *For transition kernel $q_\phi(\mathbf{y}_s \mid \mathbf{y}_0)$, we can compute upper bound the model likelihood at time $0$ as follows, for any $\epsilon > 0$*

$$\log p_\theta(x) \geq \mathbb{E}_{q_\phi(\mathbf{y}_0^v \mid x)} \mathbb{E}_{q_\phi(\mathbf{y}_\epsilon \mid \mathbf{y}_0)} \left[\log \frac{p_\theta(\mathbf{y}_0 \mid \mathbf{y}_\epsilon)}{q_\phi(\mathbf{y}_\epsilon \mid \mathbf{y}_0)} + \mathcal{L}_{mdm}(\mathbf{y}_\epsilon, \epsilon) - \log q_\phi(\mathbf{y}_0^v \mid x)\right], \qquad (57)$$

*where $\mathcal{L}_{mdm}(\mathbf{y}_\epsilon, \epsilon)$ is defined as*

$$\mathcal{L}_{mdm}(\mathbf{y}_\epsilon, \epsilon) = \mathbb{E}_{q_\phi(\mathbf{y}_{>\epsilon} \mid \mathbf{y}_\epsilon)} \left[\log \pi_\theta(\mathbf{y}_T) - \int_\epsilon^T \frac{1}{2} \|s_\phi\|_{g_\phi}^2 - \frac{1}{2} \|s_\theta - s_\phi\|_{g_\phi}^2 + \nabla \cdot f_\phi\right].$$

*Proof.* For transition kernel $q_\phi(\mathbf{y}_s \mid \mathbf{y}_0)$, we can compute upper bound the model likelihood at time $0$ following an application of the variational bound

$$\begin{aligned}
\log p_\theta(x) &= \log \int_{v_0} p_\theta(\mathbf{y}_0 = [x, v_0]) dv_0 \\
&= \log \int_{v_0, \mathbf{y}_\epsilon} p_\theta(\mathbf{y}_0, \mathbf{y}_\epsilon) dv_0 d\mathbf{y}_\epsilon \\
&= \log \int_{v_0, \mathbf{y}_\epsilon} q_\phi(\mathbf{y}_\epsilon \mid \mathbf{y}_0) \frac{q(v_0 \mid x)}{q(v_0 \mid x)} \frac{p_\theta(\mathbf{y}_0, \mathbf{y}_\epsilon)}{q_\phi(\mathbf{y}_\epsilon \mid \mathbf{y}_0)} dv_0 d\mathbf{y}_\epsilon \\
&= \log \int_{v_0, \mathbf{y}_\epsilon} q_\phi(\mathbf{y}_\epsilon \mid \mathbf{y}_0) \frac{q(v_0 \mid x)}{q(v_0 \mid x)} \frac{p_\theta(\mathbf{y}_0 \mid \mathbf{y}_\epsilon) p_\theta(\mathbf{y}_\epsilon)}{q_\phi(\mathbf{y}_\epsilon \mid \mathbf{y}_0)} dv_0 d\mathbf{y}_\epsilon \\
&\geq \mathbb{E}_{q(v_0 \mid x) q_\phi(\mathbf{y}_\epsilon \mid \mathbf{y}_0)} \left[\log \frac{p_\theta(\mathbf{y}_0 \mid \mathbf{y}_\epsilon)}{q_\phi(\mathbf{y}_\epsilon \mid \mathbf{y}_0)} - \log q_\phi(\mathbf{y}_0^v \mid x) + \log p_\theta(\mathbf{y}_\epsilon)\right]
\end{aligned}$$

A lower bound for $\log p_\theta(\mathbf{y}_\epsilon)$ can be derived in a similar manner to eq. (7), such that

$$\log p_\theta(\mathbf{y}_\epsilon) \geq \mathcal{L}_{mdm}(\mathbf{y}_\epsilon, \epsilon) = \mathbb{E}_{q_\phi(\mathbf{y}_{>\epsilon} \mid \mathbf{y}_\epsilon)} \left[\log \pi_\theta(\mathbf{y}_T) - \int_\epsilon^T \frac{1}{2} \|s_\phi\|_{g_\phi}^2 - \frac{1}{2} \|s_\theta - s_\phi\|_{g_\phi}^2 + \nabla \cdot f_\phi\right].$$

The choice of $p_\theta(\mathbf{y}_0 \mid \mathbf{y}_\epsilon)$ is arbitrary, however following Sohl-Dickstein et al. (2015); Song et al. (2021) we let $p_\theta(\mathbf{y}_0 \mid \mathbf{y}_\epsilon)$ be Gaussian with mean $\mu_{p_\theta, \epsilon}$ and covariance $\Sigma_{p_\theta, \epsilon}$. Suppose $q_\phi(\mathbf{y}_\epsilon \mid \mathbf{y}_0) = \mathcal{N}(\mathbf{y}_\epsilon \mid \mathbf{A}\mathbf{y}_0, \Sigma)$, then we select the following mean $\mu_{p_\theta, \epsilon}$ and covariance $\Sigma_{p_\theta, \epsilon}$ for $p_\theta(\mathbf{y}_0 \mid \mathbf{y}_\epsilon)$

$$\mu_{p_\theta, \epsilon} = \mathbf{A}^{-1} \Sigma s_\theta(\mathbf{y}_\epsilon, \epsilon) + \mathbf{A}^{-1} \mathbf{y}_\epsilon$$
$$\Sigma_{p_\theta, \epsilon} = \mathbf{A}^{-1} \Sigma \mathbf{A}^{-\top}$$

where $\mu_{p_\theta, \epsilon}, \Sigma_{p_\theta, \epsilon}$ are derived using Tweedie's formula (Efron, 2011) by setting $\mu_\epsilon = \mathbb{E}[\mathbf{y}_0 \mid \mathbf{y}_\epsilon]$ and $\Sigma_\epsilon = \text{Var}(\mathbf{y}_0 \mid \mathbf{y}_\epsilon)$. $\qquad \square$

We next derive this choice as an approximation of the optimal Gaussian likelihood.

### I.1 LIKELIHOOD DERIVATION

Suppose $\mathbf{y}_0 \sim q_0(\mathbf{y}_0)$ and $\mathbf{y}_\epsilon \sim \mathcal{N}(\mathbf{y}_\epsilon \mid A\mathbf{y}_0, \Sigma)$. Here, $A, \Sigma$ are the mean coefficient and covariance derived from the transition kernel at time $\epsilon$. We use Tweedie's formula to get the mean and covariance of $\mathbf{y}_0$ given $\mathbf{y}_\epsilon$ under $q$. This mean and covariance feature the true score $\nabla_{\mathbf{y}_\epsilon} \log q(\mathbf{y}_\epsilon)$. We replace the score with the score model $s_\theta$ and then set $p_\theta(\mathbf{y}_0 | \mathbf{y}_\epsilon)$ to have the resulting approximate mean and covariance. We make this choice because the optimal $p_\theta(\mathbf{y}_0 | \mathbf{y}_\epsilon)$ equals the true $q(\mathbf{y}_0 | \mathbf{y}_\epsilon)$ as discussed throughout the work.

Here $\mathbf{y}_0 = [\mathbf{x}_0, \mathbf{v}_0]$ where $\mathbf{x}_0 \sim q_{\text{data}}$.

Let $\eta$ be the natural parameter for the multivariate Gaussian likelihood $\mathcal{N}(\mathbf{y}_\epsilon \mid A\mathbf{y}_0, \Sigma)$. Then, Tweedie's formula (Efron, 2011) states that:

$$\mathbb{E}[\eta \mid \mathbf{u}_\epsilon] = \nabla_{\mathbf{y}_\epsilon} l(\mathbf{y}_\epsilon) - \nabla_{\mathbf{y}_\epsilon} l_0(\mathbf{y}_\epsilon)$$

- $l(\mathbf{y}_\epsilon) = \log q(\mathbf{y}_\epsilon)$
- $s_\theta(\mathbf{y}_\epsilon, \epsilon)$ is taken to be the true score $\nabla_{\mathbf{y}_\epsilon} \log q(\mathbf{y}_\epsilon)$ so that $\nabla_{\mathbf{y}_\epsilon} l(\mathbf{y}_\epsilon) = s_\theta(\mathbf{y}_\epsilon, \epsilon)$
- $l_0$ is the log of the base distribution defined in the exponential family parameterization.

The base distribution is a multivariate Gaussian with mean $0$ and covariance $\Sigma$, therefore $\nabla_{\mathbf{y}_\epsilon} l_0(\mathbf{y}_\epsilon) = -\Sigma^{-1}\mathbf{y}_\epsilon$,

$$\mathbb{E}[\eta \mid \mathbf{y}_\epsilon] = s_\theta(\mathbf{y}_\epsilon, \epsilon) + \Sigma^{-1}\mathbf{y}_\epsilon.$$

However, Tweedie's formula is not directly applicable since our $\mathbf{y}_\epsilon$ is not directly normal with mean $\mathbf{y}_0$. Instead, to derive the conditional mean of $\mathbf{y}_0$ given $\mathbf{y}_\epsilon$, we use the relation $\eta = \Sigma^{-1}\mathbf{A}\mathbf{y}_0$ and the linearity of conditional expectation to get

$$\begin{aligned}
\mathbb{E}[\mathbf{y}_0 \mid \mathbf{y}_\epsilon] &= \mathbb{E}[A^{-1}\Sigma\eta|\mathbf{y}_\epsilon] \\
&= A^{-1}\Sigma\mathbb{E}[\eta \mid \mathbf{y}_\epsilon] \\
&= A^{-1}\Sigma\left(s_\theta(\mathbf{y}_\epsilon, \epsilon) + \Sigma^{-1}\mathbf{y}_\epsilon\right) \\
&= A^{-1}\Sigma s_\theta(\mathbf{y}_\epsilon, \epsilon) + A^{-1}\mathbf{y}_\epsilon.
\end{aligned}$$

For the variance, we use the following relation $\mathbf{y}_\epsilon = A\mathbf{y}_0 + \sqrt{\Sigma}\epsilon$, which implies that

$$\mathbf{y}_0 = A^{-1}\mathbf{y}_\epsilon - A^{-1}\sqrt{\Sigma}\epsilon$$
$$\text{Var}\left(\mathbf{y}_0 \mid \mathbf{y}_\epsilon\right) = A^{-1}\Sigma A^{-T}.$$

Therefore, for the model posterior distribution $p_\theta(\mathbf{y}_0 \mid \mathbf{y}_\epsilon)$ we choose a Normal with mean and covariance

$$\mu_{p_\theta,\epsilon} = A^{-1}\Sigma s_\theta(\mathbf{y}_\epsilon, \epsilon) + A^{-1}\mathbf{y}_\epsilon$$
$$\Sigma_{p_\theta,\epsilon} = A^{-1}\Sigma A^{-T}$$

