# OpenReview forum: "Where to Diffuse, How to Diffuse, and How to Get Back: Automated Learning for Multivariate Diffusions"
_ICLR.cc/2023/Conference — ICLR 2023 poster_

### Official Review · Reviewer_rSRA · 2022-10-21

**Confidence:** 4
**Clarity, Quality, Novelty And Reproducibility:** Please see above.
**Correctness:** 3
**Technical Novelty And Significance:** 1
**Empirical Novelty And Significance:** 2
**Recommendation:** 3

**Strength And Weaknesses:**

## Strength:

The derivation seems to be unfied and generic. The experimental results seem good.

## Weakness:

### Novelty:

The novelty of the paper is marginal:

(1) For the theoretical derivations, all the results are readily known from previous papers (e.g., Huang et al. (2021); Durkan & Song (2021), Dockhorn et al., (2021)). The paper is just summarization of the previous derivations using a generic framework.


(2) The empirical results are not novel and lack motivation: ALDA is introduced in Mou et al. (2019). The modified version (MALDA) is not well-motivated (at least not shown in the main text) and only marginally different from the ALDA method.


### Clarity:

The paper is a ill-structured and unclear.

(1) Most of the details are deferred to the Appendix, making the paper hard to follow.

(2) The theoretical part of the paper is not structured. It is written as derivations of formula without clear goals. Assumptions are added to the derivations in the middle, e.g. Section 3.3 assumptions on $f(A,s)$; Section 3.4. linear case. Instead, the best way to present theoretical results is through a list of assumptions and theorems. This makes follow-up papers easy to verify the assumptions and use the results without knowing the details of the derivations. (Same philosophy as abstracting codes through APIs, if the authors are more familiar with coding and engineering.)

(3) The experimental part lacks motivations and exposition of the proposed method. E.g., what is the difference between ALDA and MALDA? what is the motivation for MALDA? Do you claim ALDA as your own contribution, as in the abstract and introduction section; or do you give full credit to  Mou et al. (2019)?





**Summary Of The Paper:**

The paper provides a general derivation of Evidence Lower Bounds in Multivariate Diffusion Models and some parametrization schemes. The paper also introduces MALDA diffusion model under its framework and evaluates it on MNIST and CIFAR10.

**Summary Of The Review:**

The paper is not novel and ill-structured.

---

> ### Author Response · Authors · 2022-11-19
> **Response to Reviewer rSRA (1/2)**
>
> We thank you for your insights regarding our work, and appreciate your feedback. We have made the following revisions to the work to address your comments, itemized for clarity:
>
>
> _**The paper provides a general derivation of Evidence Lower Bounds in Multivariate Diffusion Models and some parametrization schemes. The paper also introduces MALDA diffusion model under its framework and evaluates it on MNIST and CIFAR10.**_
>
>
> As mentioned in our [general comment](https://openreview.net/forum?id=osei3IzUia&noteId=6cwpbHPCu3) our technical contributions are:
> 1. Deriving ELBOs for training multivariate diffusion models (MDMs) with auxiliary variables.
> 2. Showing that the diffusion transition covariance does not need to be manually derived for
> each new diffusion. We instead demonstrate that a matrix factorization technique, previously unused in diffusion models, can automatically compute the covariance analytically
> for any linear MDM.
> 3. Using results from gradient-based MCMC to construct MDMs that exclude inference processes with poor likelihoods (i.e., a complete parameterization of inference processes
> whose stationary distribution matches the model prior).
> 4. Combining the above into an algorithm called Automatic Multivariate Diffusion Training
> (AMDT) that enables estimation without diffusion-specific derivations. AMDT enables training score models for any diffusion, including optimizing the diffusion and score jointly.
>
> For our experimental contributions, see our response below.
>
> _**For the theoretical derivations, all the results are readily known from previous papers (e.g., Huang et al. (2021); Durkan & Song (2021), Dockhorn et al., (2021)). The paper is just summarization of the previous derivations using a generic framework.**_
>
> The introduction in the revision has been revised to clarify the contributions and use of results from Huang et al. (2021); Song et al (2021).  In summary, what each of the above works does theoretically:
> 1. Huang et al (2021) and Song et al (2021) derive two ELBO’s for univariate processes (ISM and DSM)
> 2. Dockhorn et al (2021) introduce CLD, one specific multivariate diffusion, and present an objective for it. However, their objective is not an ELBO.
>
>
> In this work
> 1. We derive an ELBO for arbitrary multivariate diffusions, this requires two steps
>
>     a. Employ additional integration and bounding to lower bound the marginal distribution of the data
>
>     b. Apply Huang’s univariate bound to a process with auxiliary variables
>
>     c. We have made these points clear up front in the introduction to ensure that all underlying works are duly cited.
> 2. We use this new ELBO as an objective for MDMs. This differs from CLD’s objective and features terms that cannot be dropped when learning the forward process.
>
> Furthermore, beyond the derivation of the MDM ELBO, we make further novel theoretical contributions
>
> 1. The route existing papers take is to do analysis for a specific diffusion by solving the same set of equations from Särkkä et al (2019) [1] to solve for the transition mean and covariance. For example, CLD's derivation using Särkkä et al (2019) [1] spans over 2 full pages of heavy algebra.  Instead, using a different result from Särkkä et al (2019) [1], we show that a matrix factorization, previously unused in diffusion models, can automatically compute the covariance analytically for any linear MDM. Making use of this fact allows training diffusion models such as ALDA and MALDA without any effort, using the same 15 lines of code.
>
> 2. A key and involved step in specifying diffusion models is to check the forward process’ stationary distribution. We give the first use of the Q/D parameterization from gradient-based MCMC Ma et al 2015 [2] for diffusion based generative modeling. To make use of the Q/D parameterization, we note that the time-homogeneous result from [2] also holds for time-inhomogeneous processes. This parameterization guarantees that the stationary distribution of the inference process exists and matches the prior distribution of the generative model, thereby excluding models with poor likelihoods.
>
> 3. Finally, all of the above combined (parameterization and transition kernel automation) allow for learning the forward process in MDMs.

---

> > ### Author Response · Authors · 2022-11-19
> > **Comments to Reviewer rSRA (2/2)**
> >
> > _**The empirical results are not novel and lack motivation: ALDA is introduced in Mou et al. (2019). The modified version (MALDA) is not well-motivated (at least not shown in the main text) and only marginally different from the ALDA method.**_
> >
> > Regarding ALDA thank you for pointing this out, we understand where this confusion might have come from, while in the experiments we cite Mou et al 2019, in the introduction we did not cite them. In the revised version, we clear this confusion.
> >
> > As mentioned in the general comment [general comment](https://openreview.net/forum?id=osei3IzUia&noteId=6cwpbHPCu3) for our experimental contributions, we show that
> > 1. Existing diffusions such as ALDA, developed for MCMC sampling, can be adapted for generative modeling using the proposed algorithm AMDT.
> > 2. Trying new diffusions, like MALDA, and learning diffusions using AMDT, is easy
> > 3. For a given score model, MDMs improve bits-per-dim (BPD) compared to a univariate diffusion (VPSDE) on a number of benchmark datasets.
> > 4. Using the MDM ELBO, there are new and existing MDMs that improve performance in BPD as much as a 3 fold increase in score model size for a fixed univariate diffusion.
> >
> >
> > Regarding MALDA:
> > 1. We propose MALDA to show that our framework can be used to train a generative model for any diffusion. In particular, we made up MALDA by extending CLD from 2 to 3 dimensions.
> > 2. MALDA and ALDA, while visually similar, perform differently in all the experiments.
> > 3. The difference between MALDA and ALDA indicates the difficulty in choosing an inference diffusion for generative modeling. This is why we also learn the inference process. In the added experiments, we show that a learned inference process with 3 variables does better than MALDA.
> >
> > _**The theoretical part of the paper is not structured. It is written as derivations of formula without clear goals.**_
> >
> > We have updated the draft to present the transition kernel derivations and the MDM ELBO in a cleaner format. We have also created an algorithm AMDT to centralize the entire procedure of training MDMs in one spot.
> >
> > _**The experimental part lacks motivations and exposition of the proposed method. E.g., what is the difference between ALDA and MALDA? what is the motivation for MALDA?**_
> >
> > We propose MALDA to show that the analysis in our paper can be used to train a generative model for linear multivariate inference processes with known stationary distribution, making it easier for theoreticians and practitioners to propose and test diffusions for generative modeling.
> >
> > We made it up to show that the choice of inference matters; see ALDA vs MALDA. A priori it is difficult to know which diffusion model performs well for generative modeling, our paper allows practitioners to try many without any analysis for training.
> >
> > _**Do you claim ALDA as your own contribution, as in the abstract and introduction section; or do you give full credit to Mou et al. (2019)?**_
> >
> > Thank you for pointing out that our introduction omitted a citation to Mou et al. 2019, a reference to which we already include in our experiments. As far as the application of ALDA to diffusion modeling, this follows Dockhorn et al, which did not invent CLD but uses it for diffusion modeling. We likewise bring ALDA in as a new choice in diffusion based generative models.
> >
> >
> > _**The paper is not novel and ill-structured.**_
> >
> > Prior to our work, evaluating and prototyping diffusions was cumbersome and required pages of analysis. Our paper enables this prototyping and provides an algorithm for training multivariate diffusion models, moving beyond the 4 existing diffusions (VE-SDE, VP-SDE, sub VP-SDE, and CLD) in the literature to covering all linear inference diffusions with a fixed stationary distribution. The analysis done in our paper also enables the learning of the type of diffusion rather than just the noise scale of a specific diffusion process. The experiments show that multivariate diffusions can improve the bits-per-dim significantly without having to increase the model size.
> >
> >
> > _**Some of the paper’s claims have minor issues. A few statements are not well-supported, or require small changes to be made correct.**_
> >
> > Thank you for such a close reading of the paper. Please let us know what claims need to be corrected or are not well supported and we can address them.
> >
> >
> > [1] Särkkä, Simo, and Arno Solin. Applied stochastic differential equations. Vol. 10. Cambridge University Press, 2019.
> >
> > [2] Ma, Yi-An, Tianqi Chen, and Emily Fox. "A complete recipe for stochastic gradient MCMC." Advances in neural information processing systems 28 (2015).

---

> > > ### Comment · Reviewer_rSRA · 2022-12-03
> > > **Follow-up**
> > >
> > >
> > >
> > >
> > > After reading the authors' response and reviews from other reviewers, I decide to maintain my score, and provide a further justification for my score below.
> > >
> > > Clarity Issue No. 1:
> > >
> > > -  In Section 3.3, the authors restrict the attention to linear forward process and assume $f(y,s) = A(s)y$. In this case, the conditional distribution $q(y_s|x)$ is Gaussian. Therefore, this subsection is dedicated to derive the conditional mean and the conditional variance in terms of $A(s)$ and $g(s)$ using a matrix factorization technique.
> > >
> > > -  In Section 3.4, the authors further restrict the attention to linear forward process, invoke results from Ma et al. (2015), and state that the process indicated by $Q/D$-parametrization (13) results in a stationary distribution given by $H$, which is Gaussian in this linear case.
> > >
> > > Suggestion: I would suggest explicitly stating the required assumptions: (a). linearity & (b). Gaussian as stationary distribution, and combining section 3.3 and section 3.4. into one single section, with $Q/D$ parametrization and the differential equation of the mean and covariance under this $Q/D$ parametrization.
> > >
> > > Clarity Issue No. 2:
> > >
> > > - In the experiment section, ALDA is a scheme for accelerated gradient-based MCMC sampling method (third-order acceleration for Langevin Diffusion). I would suggest explicitly stating that ALDA has not been used in diffusion generative models before, and pointing out how ALDA differs from the previous scheme and how MALDA differs from ALDA in the main text.
> > >
> > > Additional Weakness:
> > >
> > > 1. The authors derived ELBOs for training diffusion models with auxiliary variables (Theorem 1). The theorem is generic and differs from Huang et al. (2021) only in that it includes a term $\ell_q$, which corresponds to the log-probability of the auxiliary variables. This $\ell_q$ term is standard in any ELBO derivation.
> > >
> > > 2. The authors summarize the diffusion model training in Algorithm 1, which is just a stochastic gradient descent algorithm applied to the ELBO.
> > >
> > > 3. The authors claim that their method can jointly train the parameters $\phi$ of the forward process. However, as also pointed out by Reviewer bfsX, the authors did not present any experiments on the joint training. IMHO, training $\phi$ is not trivial and requires further analysis, because the distribution for expectation of the ELBO depends on $\phi$ and it is hard to calculate a derivative w.r.t. the expectation.
> > >
> > > 4. From the Table 2, the improvements of MDMs on bits-per-dim (BPD) are not prominent: 3.106 v.s. 3.093 on CIFAR-10; 1.34 v.s. 1.31 on MNIST;

---

> > > > ### Author Response · Authors · 2022-12-04
> > > > **Response to Follow-up (1/3)**
> > > >
> > > > _Suggestion: I would suggest explicitly stating the required assumptions: (a). linearity & (b). Gaussian as stationary distribution, and combining section 3.3 and section 3.4. into one single section, with Q/D parametrization and the differential equation of the mean and covariance under this Q/D parametrization._
> > > >
> > > > Thank you for such a close reading and the suggestions regarding clarity. We have edited our local version of the draft to formally state our assumptions —linearity with a Gaussian stationary distribution ---before any of the subsections on bounds, transitions, and parameterizations.
> > > >
> > > >
> > > > _In the experiment section, ALDA is a scheme for accelerated gradient-based MCMC sampling method (third-order acceleration for Langevin Diffusion). I would suggest explicitly stating that ALDA has not been used in diffusion generative models before, and pointing out how ALDA differs from the previous scheme and how MALDA differs from ALDA in the main text._
> > > >
> > > > In the revised draft uploaded to openreview, per your earlier suggestions, we state in the introduction (page 2, second-last paragraph) that ALDA is a diffusion process that has not been used for generative modeling prior to our work. As for ALDA versus MALDA, we are glad to move their Q/D matrices to the main text. MALDA, unlike ALDA, adds noise to both auxiliary variables rather than one.
> > > >
> > > >
> > > > _The authors derived ELBOs for training diffusion models with auxiliary variables (Theorem 1). The theorem is generic and differs from Huang et al. (2021) only in that it includes a term $\ell_q$, which corresponds to the log-probability of the auxiliary variables. This $\ell_q$ term is standard in any ELBO derivation._
> > > >
> > > > Yes, our MDM ELBO differs by a term, $\ell_q$. However, it is an ELBO for a wider class of diffusion-based generative models (DBGMs) than previously studied.
> > > >
> > > > If you look at the objective used to train the only previously existing MDM, CLD, it does not include several terms necessary to make it an ELBO, including $\ell_q$. When learning the inference process, omitting these terms does not optimize a lower bound on the log likelihood of the data.
> > > >
> > > > The MDM ELBO allows us to properly evaluate a log likelihood bound for CLD. This evaluation reveals improved performance of CLD relative to what they report themselves.
> > > >
> > > >
> > > > _The authors summarize the diffusion model training in Algorithm 1, which is just a stochastic gradient descent algorithm applied to the ELBO._
> > > >
> > > > Yes, Algorithm 1 is the stochastic gradient algorithm applied to the MDM ELBO.
> > > >
> > > > 1. Given Q/D matrices, it can train a DBGM (unlike previous work which requires significant transition kernel derivations for different Q/D pairs).
> > > > 2. It can learn the Q/D matrices jointly with the score model (only explored in the univariate VP-SDE diffusion previously).
> > > >
> > > >
> > > > In summary, the algorithm’s contribution *is* that it is just an SGD algorithm which requires *significantly less effort* for training a DBGM than prior works on MDMs.
> > > >
> > > > This is in the spirit of a long line of work in variational inference (VI). For example ADVI [1] combines reparameterization gradients [2] and probabilistic programming [3] to derive automatic inference algorithms. [4] and [5] use the GAN likelihood ratio trick to derive stochastic gradient algorithms to optimize an ELBO for model-inference pairs whose sampling process is known, but have intractable likelihoods. These works all expand the class of model-inference pairs that are easy to work with. Our work is an instance of this line of work for DBGMs.
> > > >
> > > >
> > > > _References_
> > > >
> > > > [1] Kucukelbir, Alp, et al. "Automatic differentiation variational inference." Journal of machine learning research (2017).
> > > >
> > > >
> > > > [2] Kingma, Diederik P and Welling, Max. “Auto-encoding variational Bayes”. International Conference on Learning Representations, 2014.
> > > >
> > > > [3] Carpenter, Bob, et al. "Stan: A probabilistic programming language." Journal of statistical software 76.1 (2017).
> > > >
> > > > [4] Mescheder, Lars, Sebastian Nowozin, and Andreas Geiger. "Adversarial variational bayes:
> > > > Unifying variational autoencoders and generative adversarial networks." International conference
> > > > on machine learning. PMLR, 2017.
> > > >
> > > >
> > > > [5] Tran, Dustin, Rajesh Ranganath, and David Blei. "Hierarchical implicit models and likelihood-free variational inference." Advances in Neural Information Processing Systems 30 (2017).

---

> > > > > ### Author Response · Authors · 2022-12-04
> > > > > **Response to Follow-up (2/3)**
> > > > >
> > > > >  _IMHO, training $\phi$ is not trivial and requires further analysis, because the distribution for expectation of the ELBO depends on $\phi$ and it is hard to calculate a derivative w.r.t. the expectation._
> > > > >
> > > > >
> > > > > In both versions of the submission, in Table 1 & 2 and figure 1, we present results for a
> > > > > diffusion called “Learned”. This refers to the experiment where the inference process with
> > > > > parameter $\phi$ was optimized jointly with the score model.
> > > > >
> > > > > Regarding calculating the derivative of the ELBO, the objective is an expectation with respect to the Gaussian transition kernel, which contains the parameter $\phi$. The standard tool for differentiating with respect to the parameters that appear in Gaussian expectations is the reparameterization gradient. This tool goes at least as far back in VI as [1] and is used as a standard tool in machine learning [2, 3, 4, 5, 6]. Training with reparameterization gradients is as simple as instantiating a Gaussian distribution in pytorch and calling “rsample” when generating samples.
> > > > >
> > > > >
> > > > > Below we demonstrate how reparameterization works for MDM ELBOs
> > > > >
> > > > > Let
> > > > >
> > > > >  * $ m(x,s,\phi) = E_{q_\phi}[y_s|x]$
> > > > >  * $C(s, \phi)=Cov_{q_\phi}(y_s|x)$.
> > > > >  * $L(s,\phi) = Cholesky(C(s,\phi))$.
> > > > >
> > > > > Define $r(s, \phi, \epsilon) = m(x, s, \phi) + L(s, \phi) \epsilon$. Then the random variable $r(s, \phi, \epsilon)$ for $\epsilon \sim \mathcal{N}(0,I)$ has the same distribution as $y_s \sim q_\phi(y_s | x) $.
> > > > >
> > > > > For time 0, $y_0=[x , v_0]$ where $v_0 \sim q_{\phi,0}(v_0|x)$ which we choose as $\mathcal{N}(0, \gamma^2 I)$ for some $\gamma > 0$.
> > > > >
> > > > > This means we can re-write the terms in :
> > > > > $$
> > > > > E_{q_\phi(y_T|x)}\left[\log \pi(y_T)\right] - \int_0^T E_{q_\phi(y_s|x)}\left[ ||s_\theta(y_s,s)||^2_{g_\phi^2(s)} + \nabla \cdot g_\phi^2(s)s_\theta(y_s,s) - \nabla \cdot f_\phi(y_s, s) \right] ds - E_{q_\phi(y_0|x)}\left[ \log q(y_0|x) \right]
> > > > > $$
> > > > > with $\epsilon \sim \mathcal{N}(0,I)$ as
> > > > >
> > > > > - $E_\epsilon[\log \pi_\theta \left(r(T, \phi, \epsilon) \right)]$
> > > > > - $ \int_0^T E_{\epsilon}[ ||s_\theta(r(s, \phi, \epsilon) , s)||^2_{g_\phi^2(s)} + \nabla \cdot g^2_\phi(s)s_\theta( r(s, \phi, \epsilon) , s ) - \nabla \cdot f_\phi(r(s, \phi, \epsilon) , s) ] ds$
> > > > > - $E_{\epsilon}\left[ \log q_{\phi,0}([x, \gamma \epsilon ])\right]$
> > > > >
> > > > >
> > > > >
> > > > > With this reparameterization, the MDM ELBO no longer takes an expectation with respect to a distribution parameterized by $\phi$. The terms that depend on $\phi$ are $f_\phi, g_\phi, r(s, \phi, \epsilon)$ and appear only inside expectations, just like the score model parameters. Thus, auto-differentiating handles both the score model and inference process parameters in the same way.
> > > > >
> > > > > Because reparameterization is standard in the VI literature, we do not discuss this step in detail in the main text. However, if the reviewers think this is useful to add, we are glad to do so.
> > > > >
> > > > > _References_
> > > > >
> > > > > [1] Kingma, Diederik P and Welling, Max. “Auto-encoding variational Bayes”. International Conference on Learning Representations, 2014.
> > > > >
> > > > > [2] Kucukelbir, Alp, et al. "Automatic differentiation variational inference." Journal of machine learning research, 2017.
> > > > >
> > > > > [3] Vahdat, Arash, and Jan Kautz. "NVAE: A deep hierarchical variational autoencoder." Advances in Neural Information Processing Systems 33 (2020): 19667-19679.
> > > > >
> > > > > [4] Sønderby, Casper Kaae, et al. "Ladder variational autoencoders." Advances in neural information processing systems 29 (2016).
> > > > >
> > > > > [5] Haarnoja, Tuomas, et al. "Soft actor-critic: Off-policy maximum entropy deep reinforcement learning with a stochastic actor." International conference on machine learning. PMLR, 2018.
> > > > >
> > > > > [6] Mohamed, Shakir, et al. "Monte Carlo Gradient Estimation in Machine Learning." Journal of machine learning research, (2020)
> > > > >
> > > > >
> > > > >
> > > > > _The authors claim that their method can jointly train the parameters \phi of the forward process. However, as also pointed out by Reviewer bfsX, the authors did not present any experiments on the joint training._
> > > > >
> > > > > We believe there may be a misunderstanding. As mentioned above, the rows in tables 1 and 2 titled “Learned” refer to experiments where the inference process and score model are jointly learned.
> > > > >
> > > > > We believe that reviewer bfsX understood that we provide a learnable inference and generative process. Quoting reviewer bfsX here:
> > > > >
> > > > > - “Towards this end, the authors provide a new learnable inference and generative process that generalizes many familiar diffusion SDEs (e.g. VP) and provides a modified ELBO required for training.“

---

> > > > > > ### Author Response · Authors · 2022-12-04
> > > > > > **Response to Follow-up (3/3)**
> > > > > >
> > > > > > _From Table 2, the improvements of MDMs on bits-per-dim (BPD) are not prominent: 3.106 v.s. 3.093 on CIFAR-10; 1.34 v.s. 1.31 on MNIST;_
> > > > > >
> > > > > >
> > > > > > The 3.106 and 3.093 refer to CLD and the Learned process with jointly trained $\phi$. We do not emphasize improvement in BPD over CLD as a crucial take-away of the experiment. Instead:
> > > > > >
> > > > > > Our work tells us something new about an existing MDM:
> > > > > > - CLD [1] reports a BPD using an ODE model rather than directly evaluating their diffusion-based generative model. Their ODE-based BPD (3.31 BPD) is worse than the BPD we report for CLD (3.106). This improvement in performance was revealed using the MDM ELBO derived in this paper.
> > > > > >
> > > > > > Our work shows the importance of MDMs over non-MDMs:
> > > > > >
> > > > > > - CLD and Learned are both MDMs. These are trained and evaluated using the MDM ELBO and automatic kernels we describe in this work. Both of these are drastic improvements in BPDs over those reported by non-MDM DBGMs, like VPSDE, even when VPSDE is trained with a 3x larger score model.
> > > > > >
> > > > > > Our work shows that learning can supplant laborious design and derivation:
> > > > > >
> > > > > > - While CLD devotes a whole paper to carefully designing an inference process, and deriving algorithms to train it, we can just learn an inference process that performs at least as well. The fact that Learned matches CLD is a positive thing shown by our work.
> > > > > >
> > > > > > The take-aways are similar for MNIST.
> > > > > >
> > > > > > _References_
> > > > > >
> > > > > > [1] Dockhorn, Tim, Arash Vahdat, and Karsten Kreis. "Score-based generative modeling with critically-damped langevin diffusion." International Conference on Learning Representations, 2022.
> > > > > >
> > > > > >
> > > > > > _SUMMARY_
> > > > > >
> > > > > > We have provided the following clarifications
> > > > > > - The MDM ELBO includes all relevant terms for proper evaluation and training of MDMs. This has not been characterized previously.
> > > > > > - Algorithm 1 being simple is the point of our work rather than a shortcoming of our work. Our work enables training diffusion-based generative models (DBGMs) without requiring diffusion specific analysis when the inference process is linear with a Gaussian stationary distribution.
> > > > > > - The original draft and the revision present results for learned inference process parameters $\phi$ (titled “*Learned*” in the tables and figure 1). We are glad to add the above exposition on reparameterization if viewed important by the reviewers
> > > > > > - We have updated our local version of the draft for clarity by adding a clear “Assumption 1” in the top of the main technical section about linearity and gaussianity.
> > > > > >
> > > > > > We would be glad for the reviewer to consider raising their score.
> > > > > >
> > > > > > Thank you,
> > > > > > Authors

---

### Official Review · Reviewer_bfsX · 2022-10-23

**Confidence:** 3
**Correctness:** 3
**Technical Novelty And Significance:** 3
**Empirical Novelty And Significance:** 2
**Recommendation:** 8

**Clarity, Quality, Novelty And Reproducibility:**

As I mentioned above the paper is exceptionally well-written and polished. The quality of the presented theory is very high and appears to be thorough based on a quick skim of the appendix. With regards to novelty, the paper is certainly novel but leans on similar themes in past ML research in using auxiliary variables to improve modeling flexibility.

Minor:
- minor grammatical errors throughout (e.g. "by compute objectives" in the hybrid score matching section)

**Strength And Weaknesses:**

First, the paper is exceptionally well written, and the main ideas and motivation are clear and convincing. I really found the idea to include auxiliary variables as a compelling step forward given the numerous examples previously in machine learning (e.g. Augmented normalizing flows). I was initially concerned that having $K$ variables per dimension might incur a high computational cost but table 1 suggests otherwise.

From a theory standpoint, I must admit I did not have time to go through the details in the appendix but the items presented in the main paper appeared sound. The main insights of using linear diffusion and computing the transition kernels (eq 15-16) felt a bit opaque to me in the main text but I understand this is fleshed out more in Appendix C. It would be nice if the authors could move some of this material to the main text as well to help readability. The fixed inference parameterization and its extension to a learnable parameterization sounds reasonable, but as a question it appears that we get this for free? So are all MDM models have learnable inference processes by construction? It would be nice to see an ablation on this as table 2 is not very clear on this point.

I did not find many clear weaknesses in this paper. But I will outline some areas of improvement. First tables 2 and 3 can really be combined into one table, the results are identical for MDMs. Secondly, the experiments are a bit limited as they only include MNIST and CIFAR10. Most diffusion papers also include a larger dataset in Imagenet 32 X 32 or Imagenet 64 X 64. It would be good to see if MDMs are equally performant in this setting. Also, it would be nice to get a convergence vs. timestep analysis on ELBO for MDMs vs other diffusion models. Do MDMs converge faster due to higher modeling flexibility? Finally, it appears ALDA is significantly worse than MALDA? The intuition for this is not very clear to me, can the authors provide more details here?



**Summary Of The Paper:**

This paper introduces Multivariate Diffusion Models which extend standard diffusion models to include auxiliary variables. Towards this end, the authors provide a new learnable inference and generative process that generalizes many familiar diffusion SDEs (e.g. VP) and provides a modified ELBO required for training. The authors also introduce two linear diffusion models in ALDA and MALDA and show that these lead to state-of-the-art BPD and NLL on MNIST and CIFAR10.

**Summary Of The Review:**

Overall this paper is of high quality and has a nice theoretical component. The experiments are limited to small datasets but there does not appear to be a significant bottleneck in scaling them. Based on this I recommend accepting this paper with a score 8.

---

> ### Author Response · Authors · 2022-11-19
> **Comment to Reviewer bfsX**
>
> We are glad you like the work and find the exposition of its unique points clear. We are happy to provide more details to your comments.
>
>
> _**The main insights of using linear diffusion and computing the transition kernels (eq 15-16) felt a bit opaque to me in the main text but I understand this is fleshed out more in Appendix C. It would be nice if the authors could move some of this material to the main text as well to help readability.**_
>
> Thank you for this suggestion, we have moved some details from appendix C to the main text.
>
>
> _**The fixed inference parameterization and its extension to a learnable parameterization sounds reasonable, but as a question it appears that we get this for free? So are all MDM models have learnable inference processes by construction? It would be nice to see an ablation on this as table 2 is not very clear on this point.**_
>
> Yes, the learnable parameterization comes for free with the derivation of the MDM ELBO and the automation of its estimation and optimization. Table 2 keeps the inference diffusion fixed. We are happy to address any further questions.
>
> _**The experiments are a bit limited as they only include MNIST and CIFAR10. Most diffusion papers also include a larger dataset in Imagenet 32 X 32 or Imagenet 64 X 64. It would be good to see if MDMs are equally performant in this setting.**_
>
> We have added numbers for Imagenet32 in the revised version of the dataset. The experiments demonstrate that adding auxiliary variables and learning the inference diffusion improves the ELBO. Due to limited compute, we do not evaluate ALDA or learned a diffusion with 3 variables on ImageNet32.
>
>
> _**Also, it would be nice to get a convergence vs. timestep analysis on ELBO for MDMs vs other diffusion models. Do MDMs converge faster due to higher modeling flexibility?**_
>
> This is a great question. In general, MDMs increase the ELBO faster, but it depends on the quality of the MDM. For instance, ALDA performs worse than VPSDE in all our experiments. Also, we observe that for a fixed number of auxiliary variables, learning typically increases speed of convergence.
>
>
> _**Finally, it appears ALDA is significantly worse than MALDA? The intuition for this is not very clear to me, can the authors provide more details here?**_
>
> ALDA is a diffusion process used for gradient-based MCMC, motivated similarly as CLD. We have empirically seen that at smaller times ALDA’s transition covariance is closer to singular, which penalizes the score matching part of the loss, making optimization harder.
>
> We introduce MALDA to highlight that a priori it is difficult to understand which diffusion process would work well for generative modeling, and that’s why we advocate learning it. We also added experiments with a learned inference process with 3 variables, which we saw performed better than MALDA for both CIFAR and MNIST.
>
> _**With regards to novelty, the paper is certainly novel but leans on similar themes in past ML research in using auxiliary variables to improve modeling flexibility.**_
>
> Thanks for pointing this out, auxiliary variable models improve performance for a number of generative models such as normalizing flows, Hamiltonian Monte Carlo, variational inference. Motivated by CLD, we extend the use of MDMs beyond one specific diffusion process.
>
>
> _**Minor grammatical errors throughout (e.g. "by compute objectives" in the hybrid score matching section)**_
>
> Thanks so much for your super close reading of the paper. We have tried to catch as many of these as possible.

---

### Official Review · Reviewer_yDwB · 2022-10-25

**Confidence:** 3
**Correctness:** 4
**Technical Novelty And Significance:** 4
**Empirical Novelty And Significance:** 3
**Recommendation:** 8

**Clarity, Quality, Novelty And Reproducibility:**

The paper is clearly written and the technical part is of high quality. The novelty is good. The experiments seem reproducible.

**Strength And Weaknesses:**

Strength:

- The paper is well-written and introduces an interesting novel approach for establishing diffusion probabilistic models in auxiliary space.
- The motivation of the work is explained very well in Section 1 and Section 4, which is helpful for understanding the paper.
- Technical contributions are solid (to the best of my knowledge).
- Experimental results achieve state-of-the-art likelihood (in terms of bits-per-dim).

Questions:
- How is the auxiliary variable initialized in time zero? Is it sampled from $q_\phi(\mathbf{y}^v_0|x)$ during training? It seems that $q_\phi(\mathbf{y}^v_0|x)$ can be any distribution in theory. In the paper, an example of $q_\phi(\mathbf{y}^v_0|x)=\mathcal{N}(\mathbf{0},\mathbf{I})$ is given, but this could be quite different from the data distribution, i.e., $q_\phi(\mathbf{y}^z_0|x)$. Since in the practical implementation the score for each dimension is output from different channels from the same model, I wonder whether a better choice of $q_\phi(\mathbf{y}^v_0|x)$ could alleviate the training burden and perhaps results in a better likelihood or faster training?

Minors:
- In Eq.(18) no text explanation under the underbrace.
- On page 6 there is something wrong with the typography.


**Summary Of The Paper:**

This paper proposes to improve diffusion probabilistic models by considering multivariate diffusions. The motivation inherits previous probabilistic modeling methods that use auxiliary variables. By augmenting the diffusion space, it enables mixing between data dimensions and auxiliary dimensions, leading to better-aligned generative and inference processes. Based on this the paper further demonstrates how to parameterize the inference process, e.g., MALDA. Experimental results are promising in terms of test likelihood estimation.

**Summary Of The Review:**

Overall, I think this paper is of good quality, in that it presents a novel and interesting perspective of understanding and a concrete approach for diffusion models, achieving good results on standard benchmarks.

---

> ### Author Response · Authors · 2022-11-19
> **Response to Reviewer yDwB**
>
> We are glad you like the work and find the exposition of its points clear. We are happy to provide more details to your questions.
>
>
> _**How is the auxiliary variable initialized in time zero? .... I wonder whether a better choice of
> q(v0|x) could alleviate the training burden and perhaps results in a better likelihood or faster training?**_
>
>
> This is a great question:
> 1. The implication of having $q_{\phi}(v_0|x)$ is that $v_0$ and x are dependent. This dependence provides more flexibility in where the auxiliary distribution in the generative model needs to converge.
> 2. With a distribution $q(v_0)$ that is independent of the data, the model has to make the auxiliary variables independent of the data as generation completes (time goes to 1), after using them at some intermediate point in the generative process.
> 3. Which one of these two choices is ultimately better for modeling the marginal data distribution is an open question. There are several possibilities worth trying, some of them being the initial auxiliary variable distribution can be
>
>     a. Equal to the data
>
>     b. Gaussian with mean equal to the data or other random transformations of the data
>
>     c. The MDM ELBO permits learning of $q(v_0|x)$, though it’s form should be simple to ensure computational feasibility.

---

### Comment · Area_Chair_RCHM · 2022-11-15
**Rebuttal**

Dear authors,

A rebuttal to the reviews would be highly appreciated.

Kind regards,
Your AC

---

### Author Response · Authors · 2022-11-19
**Overview and summary of changes**

We would like to thank the area chair and the reviewers for their time and comments on the paper.

We are glad the reviewers found our work to be interesting and well-motivated.
1. “I really found the idea to include auxiliary variables as a compelling step forward given the numerous examples previously in machine learning (e.g. Augmented normalizing flows).” Reviewer bfsX
2. “The paper is well-written and introduces an interesting novel approach for establishing diffusion probabilistic models in auxiliary space.”. Reviewer yDwB
3. “Overall this paper is of high quality and has a nice theoretical component.” Reviewer bfsX

Using the existing mode of analysis, experimenting with multivariate diffusions would be a cumbersome process as each new diffusion used in inference requires significant diffusion specific analysis for deriving the stationary distribution, transition density. Further no ELBO has been derived for multivariate diffusions, making it unclear how to directly evaluate these diffusion models or how to optimize the inference diffusion.

Our work derives an ELBO for multivariate diffusions and automates the derivations required for estimating and optimizing the ELBO for linear multivariate diffusions with a stationary distribution. We put these steps into an algorithm, automated multivariate diffusion training (AMDT), that allows for easy and rapid prototyping of new inference diffusions. To experiment with a new inference diffusion, a practitioner needs to just specify the inference diffusion parameters.

A priori it is difficult to handcraft diffusions that work well on any dataset and any score model. AMDT enables optimizing over linear multivariate diffusions with a stationary distribution, freeing a practitioner from such choices.

In summary, our technical contributions are:
1. Deriving ELBOs for training multivariate diffusion models (MDMs) with auxiliary variables.
2. Showing that the diffusion transition covariance does not need to be manually derived for
each new diffusion. We instead demonstrate that a matrix factorization technique, previously unused in diffusion models, can automatically compute the covariance analytically
for any linear MDM.
3. Using results from gradient-based MCMC to construct MDMs that exclude inference processes with poor likelihoods (i.e., a complete parameterization of inference processes
whose stationary distribution matches the model prior).
4. Combining the above into an algorithm called Automatic Multivariate Diffusion Training
(AMDT) that enables estimation without diffusion-specific derivations. AMDT enables training score models for any diffusion, including optimizing the diffusion and score jointly.

As for our experimental contributions, we show that:
1. Existing diffusions such as ALDA, developed for MCMC sampling, can be adapted for generative modeling using the proposed algorithm AMDT.
2. Trying new diffusions, like MALDA, and learning diffusions using AMDT, is easy
3. For a given score model, MDMs improve bits-per-dim (BPD) compared to a univariate diffusion (VPSDE) on a number of benchmark datasets.
4. Using the MDM ELBO, there are new and existing MDMs that improve performance in BPD as much as a 3 fold increase in score model size for a fixed univariate diffusion.

_Note on evaluation metric_:

For our evaluations, we use the evaluation standard followed by several repositories for diffusion models and evaluation methods in published papers on diffusion models [1] .  This evaluation was based on ISM, which we found to have high variance estimates of the evaluation metric, the ELBO.

To account for this variance, we evaluate this on importance weighted DSM with one thousand Monte Carlo samples, matching the most extensive evaluation done of continuous-time diffusion lower bounds [2]. We have updated the evaluations in the revised draft.

**Summary of changes to paper in response to reviewers**

1. We updated the introduction to make clear the methodological and experimental contributions to address the concerns of reviewer rSRA.
2. We have added a citation that was later in the paper to the introduction (reviewer rSRA).
3. We have updated the section on the transition kernel automation to provide more intuition about the derivation for the transition mean and covariance (reviewer bfsX).
4. We also clarify the presentation of the ELBO for multivariate diffusion models (reviewer rSRA).
5. We also added experiments with the ImageNet32 dataset (reviewer bfsX).


We plan to release the code soon, but are glad to provide it if the reviewers find it helpful.

_References_

[1] Chin-Wei Huang, Jae Hyun Lim, Aaron Courville. A Variational Perspective on Diffusion-Based Generative Models and Score Matching. 2021

[2] Yang Song, Conor Durkan, Iain Murray, Stefano Ermon. Maximum Likelihood Training of Score-Based Diffusion Models. 2021

---

### Decision · Program_Chairs · 2023-01-20

**Decision:**

Accept: poster

**Justification For Why Not Higher Score:**

The empirical section is somewhat weak.

**Justification For Why Not Lower Score:**

The strong theoretical section makes up for weaknesses in the empirical section.

**Metareview: Summary, Strengths And Weaknesses:**

Ratings: 8/8/3.
Confidences: 3/3/4.

In this work, the authors study the (fully?) general case of diffusion models, where the forward diffusion process and its reverse are parameterized by non-diagional covariances for the diffusion term; i.e. multivariate diffusion models (MDMs). In particular, the authors look at the case where auxiliary dimensions are included. Their contributions are (1) deriving ELBOs for training multivariate diffusion models with auxiliary variables, (2) Showing that the diffusion transition covariance does not need to be manually derived for each new diffusion, (3) construct a family of diffusion models whose q(z_T) is a spherical Gaussian, (4) combining the above into an algorithm called Automatic Multivariate Diffusion Training (AMDT) that enables estimation without diffusion-specific derivations. In experiments, the authors show that these innovations with multivariate diffusions provide some performance gain.

This appears to be a theoretically strong paper. Neither me nor the reviewers could find major problems with the main results. I think the main results could have significant impacts on the community.

However, some reviewers were unhappy about the presentation. The empirical section also has some weaknesses. The experiments are relatively small-scale, and the gains in likelihood are quite small. In addition, I would have liked to see stronger baselines. For example, the authors use a diffusion model baseline that gets ~3.1 BPD on CIFAR-10, which is far from SOTA; a more logical choice would have been Variational Diffusion Models [Kingma et al, 2021] which gets 2.65 BPD on the same dataset (whose code is available). Why do the authors choose such a sub-par baseline? Most importantly, it's currently not completely clear if the proposed methods can help at all getting better likelihoods in such a regime.

The authors provided a rebuttal to all reviews, but only bfsX engaged, which also gave the only reject rating in the end. However, we feel that the authors did a good job with his rebuttal. Ultimately, we think this outweighed by the positive reviews, and the positive reading by myself.


**Note From Pc:**

if the above contains the word "oral" or "spotlight" please see: "oral" presentation means -> notable-top-5% and "spotlight" means -> notable-top-25%. As stated in our emails, we are disassociating presentation type from AC recommendations

**Summary Of Ac-Reviewer Meeting:**

Unfortunately, the reviewers were not responsive, even after reminders, so a AC/Reviewer meeting could not be scheduled.